# MSpecTmol: A Multi-Modal Spectroscopic Learning Framework for Molecular Structure Identification

## Abstract

Spectroscopic techniques are indispensable for the elucidation of molecular structures, particularly for novel molecules with unknown configurations. However, a fundamental limitation of any single spectroscopic modality is that it provides an inherently circumscribed and fragmented view, capturing only specific facets of the complete molecular structure, which is often insufficient for unequivocal and robust characterization. Consequently, the integration of data from multiple spectroscopic sources is imperative to overcome these intrinsic limitations and achieve a comprehensive and accurate structural characterization. In this work, we introduce **MSpecTmol**, a novel **M**ulti-modal **Spec**trum information fusion learning framework for **Mol**ecule structure elucidation. By extending information bottleneck theory, our framework provides a principled and adaptive approach to fusing spectra. It designates a primary modality to extract core molecular features while leveraging auxiliary inputs to enrich the representation. To validate the end-to-end effectiveness of our framework, we design a two-fold evaluation: molecular substructure classification to probe its discriminative power in identifying substructures, and extends this knowledge to reconstruct plausible 3D structures. Our results not only demonstrate state-of-the-art performance in molecular substructure classification but also achieve near-experimental accuracy (˜0.68Å) in molecular conformation reconstruction. These findings underscore the model's capacity to learn interpretable features aligned with chemical intuition, thereby paving the way for future advances in automated and reliable spectroscopic analysis. Our code can be found at https://anonymous.4open.science.

## 1 Introduction

The rapid advancements in artificial intelligence (AI) have significantly propelled research in the chemical sciences Goh et al. (2017); Divya et al. (2024); Rial (2024); Ananikov (2024), enabling breakthroughs in molecular property prediction Feinberg et al. (2018); Walters & Barzilay (2020), drug design Blundell (1996); Riccardi et al. (2018), and drug-drug interaction studies Zhao et al. (2024); Wang et al. (2024). AI not only achieves high-precision predictions without compromising accuracy but also enhances trust in its applications through interpretable models Chander et al. (2024); Rane et al. (2024). These developments have increasingly integrated AI into chemistry as an indispensable tool. Notably, the vast majority of existing studies are *post-designed*, meaning that they operate on molecules with known structures, represented either as molecular graphs or SMILES strings Du et al. (2023); Xia et al. (2023).

However, for a novel, unknown molecule, chemists must first determine its fundamental structure before exploring its properties Hastings et al. (2021); Stanzione et al. (2021). In such cases, spectroscopic techniques serve as powerful tools for structural determination, fundamentally projecting high-dimensional chemical structures into lower-dimensional spectral representations as present in Figure 1 Barone et al. (2021); Meza Ramirez et al. (2021). Spectral techniques such as nuclear magnetic resonance (NMR), infrared (IR) spectroscopy, and mass spectrometry (MS) could provide critical insights into molecular structures Fontana & Widmalm (2023); Manogaran et al. (2024), including the presence or absence of functional groups Ge et al. (2021) which plays a crucial role in confirming structural assignments and ensuring the reliability of downstream chemical analysis.

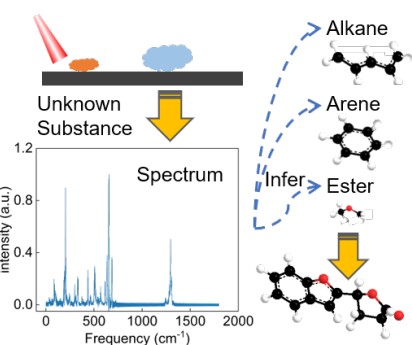

Figure 1: Schematic representation of inferring an unknown substance using spectral analysis techniques.

Yet, the inherent limitations of individual spectroscopic methods, due to their low-dimensional characteristics and the restricted information they contain Xue et al. (2024); Bose et al. (2021), necessitate the integration of multiple spectroscopic sources to achieve more precise molecular determination. For instance, IR spectroscopy focuses on functional group vibrations, ultraviolet-visible (UV-Vis) spectroscopy reflects overall molecular properties, and NMR provides information about the local atomic environment Chen et al. (2023); Manogaran et al. (2024). Each spectroscopic modality encapsulates distinct representational features and operates within different physically meaningful ranges Barone et al. (2021). Therefore, a key challenge lies in fully leveraging the available spectroscopic data to extract its physical significance and enable accurate molecular structural determination Meza Ramirez et al. (2021).

In this work, we propose a novel **M**ulti-modal **SpecT**rum information fusion learning framework based on information bottleneck theory for **Mol**ecule confirmation, termed **MSpecTmol**, to integrate multi-modal spectroscopic data. Our framework adopts a primary-auxiliary synergistic modeling approach, where the roles of primary and auxiliary representations are clearly delineated. By extending the multi-objective information bottleneck theory to this setting, we enable the primary modality to capture core information by filtering out redundant or irrelevant features, while the auxiliary modalities supplement the primary representation to enhance and refine the results. To comprehensively validate our framework's end-to-end effectiveness across the entire spectrum-to-molecule workflow, we rigorously applied it to two critically important tasks: molecular identification and spectrum-conditioned molecular conformation generation. In the molecular identification task, MSpecTmol significantly outperformed state-of-the-art baseline methods across both simulated and experimental spectra, achieving an F1-score of 0.959. Furthermore, the framework similarly demonstrated its capabilities for intricate structural elucidation in the challenging spectrum-conditioned conformation generation task, achieving an average RMSD of 0.682 Å. Moreover, MSpecTmol captures critical spectroscopic fragments that align well with chemical intuition, providing a degree of interpretability for its predictions. The synergy between primary and auxiliary modalities offers a flexible strategy for researchers to adapt to various chemical challenges, further improving performance outcomes. We envision that molecular identification through spectroscopic data will become a key research focus in automated laboratory workflows. MSpecTmol represents a promising solution to this challenge, offering a robust and interpretable framework for this.

## 2 METHODOLOGY

In this section, we introduce our proposed framework, called **MSpecTmol**, a novel *multi-modal information fusion learning* framework that refines representations based on the distinct roles of the underlying information. First, we formally define **MSpecTmol** (Section 2.1). Next, we present the overall model architecture (Section 2.2), followed by the final optimization process (Section 2.3).

### 2.1 PRIMARY-AUXILIARY INFORMATION BOTTLENECK

In this work, we focus on learning the core representations $T_m$ and $T_a$ from the input primary spectrum $X_m$ and auxiliary spectra $X_a$.

*Primary-Auxiliary Information Bottleneck. (PA-IB)* Given the primary spectrum variables $X_m$, the auxiliary spectra variables $X_a$, and the target variable $Y$, the Primary-Auxiliary Information Bottleneck theory aims to compress $X_m$ into a bottleneck variable $T_m$ while preserving the information needed to predict $Y$, and to compress $X_a$ into a bottleneck variable $T_a$ while preserving the information needed to predict $Y$ conditioned on $X_m$. Formally, we seek to solve:

$$\min -I(Y; T_m) - I(Y; T_a \mid T_m) + \alpha\, I(X_m; T_m) + \beta\, I(T_a; X_m, X_a), \tag{1}$$

where $\alpha$ and $\beta$ are Lagrange multipliers that balance the mutual information terms.

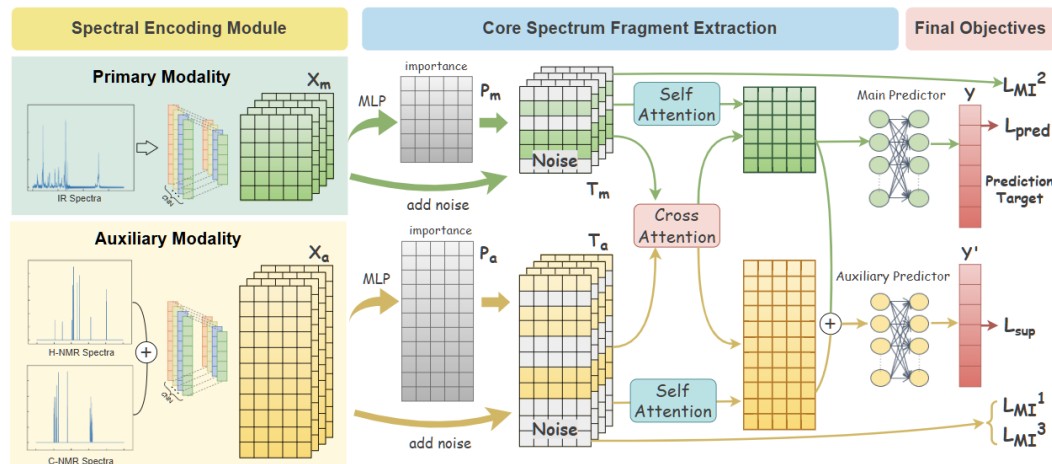

Figure 2: Illustration of the **MSpecTmol** framework. The model processes one primary and multiple auxiliary spectral modalities. Initially, vector representations are generated via linear interpolation and then fed into 1D-CNN layers to extract feature matrices. Subsequently, the core IB-Spectra module distills essential information from the primary modality and complementary features from the auxiliary inputs, producing a compact feature vector for downstream classification.

*IB-Spectra.* Given a set of spectra $(X_m, X_a)$ and its corresponding label information $\mathbf{Y}$, we identify the optimal primary spectrum $\mathcal{T}_{MIB}$ and auxiliary spectrum $\mathcal{T}_{AIB}$ under the PA-IB principle:

$$\mathcal{T}_{\text{MIB}}, \mathcal{T}_{\text{AIB}} = \underset{\mathcal{T}_{\text{MIB}}, \mathcal{T}_{\text{AIB}}}{\arg\min} \Big[-I(Y; T_m) - I(Y; T_a \mid T_m) + \alpha\, I(X_m; T_m) + \beta\, I(T_a; X_m, X_a)\Big], \tag{2}$$

This objective involves the following four components:

- $-I(Y; T_m)$: Encourages the primary representation $T_m$ to preserve the most predictive information about $Y$. This term corresponds to the classical IB objective and ensures that $T_m$ serves as the main carrier of task-relevant information.
- $-I(Y; T_a \mid T_m)$: Drives the auxiliary representation $T_a$ to complement $T_m$ by capturing additional information that is not contained in $T_m$, thus improving the overall predictive capacity.
- $\alpha\, I(X_m; T_m)$: Regularizes the complexity of $T_m$ by penalizing excessive mutual information with the input $X_m$, thereby promoting a compact and generalized encoding.
- $\beta\, I(T_a; X_m, X_a)$: Limits the complexity of $T_a$ by minimizing its mutual information with the combined inputs $(X_m, X_a)$, encouraging selective representation of auxiliary information.

Combining these four terms, the final objective is: The primary representation $T_m$ captures the core information necessary for predicting $Y$. The auxiliary representation $T_a$ complements $T_m$ by providing any additional information needed for $Y$, while maintaining low complexity.

## 2.2 MODEL ARCHITECTURE

### 2.2.1 SPECTRAL ENCODING MODULE

Here, $\{X_m, X_a\}$ denote a pair of input spectra, with $X_m$ as the *primary* spectrum and $X_a$ as the *auxiliary* spectra. To unify dimensions, each spectrum is interpolated to 600 uniformly spaced points by linear interpolation, where the point value $x$ is updated to $X'$:

$$X'(x) = X(x_i) + \frac{(x - x_i)}{(x_{i+1} - x_i)} \cdot (X(x_{i+1}) - X(x_i)), \quad x \in [x_i, x_{i+1}] \tag{3}$$

where $x_i$ and $x_{i+1}$ are consecutive points in the original spectrum, and $X(x)$ represents the updated value at $x$. This procedure is applied independently to two different spectra designated as auxiliary

inputs. After interpolation, both spectra are normalized and concatenated to create a single 1200-dimensional auxiliary spectrum, $X_a$.

The input spectra $X_m$ and $X_a$ are encoded through two stages of 1D convolutional layers, each followed by batch normalization, ReLU activation, and max pooling. The use of 1D convolutions is motivated by the sequential nature of spectral data, where capturing local patterns is essential for extracting meaningful features. Convolutional layers enable the model to automatically learn hierarchical representations and spatially invariant characteristics from the spectra.

$$O^m = \texttt{MaxPool1D}(\text{ReLU}(\text{BatchNorm}(\text{Conv1D}(X_m)))), \tag{4}$$

$$O^a = \texttt{MaxPool1D}(\text{ReLU}(\text{BatchNorm}(\text{Conv1D}(x_a)))), \tag{5}$$

Here, $O^m \in \mathbb{R}^{c \times d}$ and $O^a \in \mathbb{R}^{c \times d}$, where $c$ denotes the number of convolutional channels and $d$ represents the length after linear interpolation. The pool size is set to 2.

### 2.2.2 CORE SPECTRUM EXTRACTION

In this section, we extract core spectral segments by first transposing the frequency representations: $H^m = O_m^{(t)}$, $H^a = O_a^{(t)}$ , where $(t)$ denotes the transpose operation. For the primary spectrum, we compress $X_m$ into $T_m$ by injecting noise into its learned embedding, encouraging the model to suppress less informative frequency bands. For the auxiliary spectrum, we similarly derive $T_a$ from $X_m$, $X_a$, and $T_m$, guided by equation 7. The key idea is to enable the model to inject noise into insignificant frequency bands while injecting less noise into more informative ones Yu et al. (2022). we could calculate the probability $p_i^m$ and $p_i^a$ using an MLP, i.e.,

$$p_i^m = \text{MLP}(\mathbf{H}_i^m) \quad p_i^a = \text{MLP}(\mathbf{H}_i^m \parallel \mathbf{H}_i^a \parallel \mathbf{T}_i^m). \tag{6}$$

With the $p_i^m$ and $p_i^a$, we replace $\mathbf{H}_i^m$ and $\mathbf{H}_i^a$ of frequency band $i$ with noise $\epsilon$, i.e.,

$$\mathbf{T}_i^m = \lambda_i^m \mathbf{H}_i^m + (1 - \lambda_i^m)\epsilon^m, \quad \mathbf{T}_i^a = \lambda_i^a \mathbf{H}_i^a + (1 - \lambda_i^a)\epsilon^a, \tag{7}$$

where $\lambda_i^m \sim \text{Bernoulli}(p_i^m)$ and $\epsilon^m \sim \mathcal{N}(\mu_m, \sigma_m^2)$. Here, $\mu_m$ and $\sigma_m^2$ are mean and variance of $\mathbf{H}^m$, respectively. Thus, the information of $X_m$ is compressed into $T_m$ with the probability $p_i^m$, by replacing unimportant frequency bands with noise. Similarly, for the core auxiliary spectrum, The information from $X_m$, $T_m$, and $X_a$ is compressed into $T_a$ with the same probability $p_i^a$.

Moreover, to make the sampling process differentiable, the Gumbel-Softmax is adopted Maddison et al. (2016); Jang et al. (2016) for the discrete random variable $\lambda_i$, i.e.,

$$\lambda_i = \sigma \left( \frac{1}{t} \log \left( \frac{p_i}{1 - p_i} \right) + \log \left( \frac{u}{1 - u} \right) \right), \tag{8}$$

where $u \sim \text{Uniform}(0, 1)$, and $t$ is the temperature hyperparameter, set to 1.0 in this work. A detailed sensitivity analysis of $t$ is provided in Appendix J.

### 2.3 MODEL OPTIMIZATION

To train the model while simultaneously detecting the core primary spectra and core auxiliary spectra, we optimize the model with the objective function defined in equation 1 as follows:

$$\min -I(Y; T_m) - I\big(Y; T_a \mid T_m\big) + \alpha\, I\big(X_m; T_m\big) + \beta\, I\big(T_a; X_m, X_a\big), \tag{9}$$

where each term corresponds to prediction or compression, respectively. In the following sections, we provide the upper bounds of each term, which should be minimized during training.

### 2.3.1 MINIMIZING $-I(Y; T_m)$

**Proposition 3.1 (Upper bound of $-I(Y; T_m)$)** Given the primary spectra $X_m$, and its label information $\mathbf{Y}$, we have:

$$\begin{aligned} -I(\mathbf{Y}; T_m) &\leq \mathbb{E}_{T_m, \mathbf{Y}}[-\log p_\theta(\mathbf{Y}|T_m)] \\ &= \mathbb{E}_{(\mathbf{Y}, T_m)} \log \left[ P_\theta\left(\mathbf{Y} \mid T_m\right) \right] + H(\mathbf{Y}) := \mathcal{L}_{\text{pred}}, \end{aligned} \tag{10}$$

where $H(\mathbf{Y})$ is the entropy of the label $\mathbf{Y}$, which is constant across the dataset and can be omitted in the optimization. $p_\theta(\mathbf{Y}|T_m)$ is the variational approximation of the true posterior $p(\mathbf{Y}|T_m)$. Minimizing this upper bound corresponds to minimizing the prediction loss $\mathcal{L}_{\text{pred}}(\mathbf{Y}, T_m)$, which is modeled as the cross-entropy loss for classification. The proof can be found in Appendix F.1.1.

### 2.3.2 MINIMIZING $-I\big(Y; T_a \mid T_m\big)$

**Proposition 3.2 (Upper bound of $-I\big(Y; T_a \mid T_m\big)$)** We decompose the term using the chain rule of mutual information:

$$-I\left(Y; T_a \mid T_m\right) = -I(Y; T_a, T_m) + I(T_a; T_m)$$
$$\leq \mathbb{E}_{(\mathbf{Y}, T_a, T_m)} \log\left[P_\theta\left(\mathbf{Y} \mid T_a, T_m\right)\right] + \mathbb{E}_{t_m \sim p(t_m)}\left[\mathrm{KL}\left(p(t_a \mid t_m) \| q(t_a)\right)\right] \quad (11)$$
$$:= \mathcal{L}_{\mathrm{sup}} + \mathcal{L}_{\mathrm{MI}^1}.$$

Here, $\mathcal{L}_{\mathrm{sup}}$ represents the supervised prediction loss $\mathcal{L}_{\mathrm{pred}}(\mathbf{Y}, T_m, T_a)$, which is implemented as cross-entropy for prediction. The second term, $\mathcal{L}_{\mathrm{MI}^1}$, corresponds to the KL divergence between the posterior $p(t_a \mid t_m)$ and a prior $q(t_a)$, regularizing the relationship between the auxiliary spectra $T_a$ and primary spectra $T_m$. This divergence is minimized using variational inference, and is estimated by averaging over samples of $t_m$. Detailed derivations can be found in Appendix F.1.2. Specifically, as shown in Appendix L, we investigate the impact of different prior distributions of $q(t_m)$ and $q(t_a)$ on model performance, and select the best prior distribution as the distribution for MSpecTmol.

### 2.3.3 MINIMIZING $I\left(X_m; T_m\right)$

**Proposition 3.3 (Upper bound of $I\left(X_m; T_m\right)$)** We apply the variational approximation to bound the mutual information term:

$$I\left(X_m; T_m\right) \leq \mathbb{E}_{t_m \sim p(t_m)}\left[\mathrm{KL}\left(p(t_m \mid x_m) \| q(t_m)\right)\right] := \mathcal{L}_{\mathrm{MI}^2}. \quad (12)$$

Here, $\mathcal{L}_{\mathrm{MI}^2}$ corresponds to the KL divergence between the posterior $p(t_m \mid x_m)$ and a prior $q(t_m)$. The KL divergence is computed using variational inference and is estimated by averaging over samples of $x_m$. The detailed derivation is provided in Appendix F.1.3.

### 2.3.4 MINIMIZING $I\big(T_a; X_m, X_a\big)$

**Proposition 3.4 (Upper bound of $I\big(T_a; X_m, X_a\big)$)** We minimize the mutual information between the auxiliary spectra $T_a$ and both the primary spectra $X_m$ as well as the auxiliary spectra $X_a$:

$$I\big(T_a; X_m, X_a\big) \leq \mathbb{E}_{t_a, x_a \sim p(x_m, x_a)}\left[\mathrm{KL}\left(p(t_a \mid x_m, x_a) \| q(t_a)\right)\right] := \mathcal{L}_{\mathrm{MI}^3}. \quad (13)$$

Here, $\mathcal{L}_{\mathrm{MI}^3}$ represents the KL divergence between the posterior $p(t_a \mid x_m, x_a)$ and a prior $q(t_a)$. The KL divergence is estimated using variational inference, with derivations detailed in Appendix F.1.4.

## 2.4 FINAL OBJECTIVES

The final objective function used for training is given by:

$$\mathcal{L}_{\mathrm{total}} = \mathcal{L}_{\mathrm{sup}} + \mathcal{L}_{\mathrm{pred}} + \mathcal{L}_{\mathrm{MI}^1} + \alpha\,\mathcal{L}_{\mathrm{MI}^2} + \beta\,\mathcal{L}_{\mathrm{MI}^3} \quad (14)$$

where $\alpha$ and $\beta$ control the trade-off between prediction accuracy and compression. The detailed derivations and proofs for $\mathcal{L}_{\mathrm{pred}}, \mathcal{L}_{\mathrm{sup}}, \mathcal{L}_{\mathrm{MI}^1}, \mathcal{L}_{\mathrm{MI}^2}$, and $\mathcal{L}_{\mathrm{MI}^3}$ are provided in above.

## 3 EXPERIMENT AND ANALYSES

We present experimental results to demonstrate the effectiveness of MSpecTmol under two tasks: molecular identification and spectrum-conditioned molecular conformation generation. In this section, we conduct extensive experiments to address the following research questions:

- **RQ1:** Can MSpecTmol accurately perform fine-grained classification of molecular substructures?
- **RQ2:** Can MSpecTmol accurately generate 3D molecular conformations by spectra?
- **RQ3:** Can MSpecTmol provide interpretable insights?

## 3.1 DATASETS AND SETUPS

**Datasets.** We utilize the large-scale dataset from Alberts et al. Alberts et al. (2024) for molecular structure elucidation, which contains 794K molecules with simulated IR, $^1$H-NMR, $^{13}$C-NMR, and

Table 1: F1-scores for predicting functional groups. For multi-modal settings, the primary modality is indicated in **bold**. Baseline models are invariant to the choice of primary modality, whereas MSpecTmol leverages this information to achieve superior performance. The best results are highlighted in **bold**, and the second-best are underlined.

| Spectrum Config. | 1D-CNN | Transformer | Wu et al. | Alberts et al. | MSpecTmol |
|---|---|---|---|---|---|
| **Alberts et al. (Simulated Spectra)** | | | | | |
| IR | $\underline{0.895}_{(0.002)}$ | $0.881_{(0.021)}$ | $0.886_{(0.013)}$ | $0.891_{(0.007)}$ | $\mathbf{0.923}_{(0.004)}$ |
| $^{13}$C-NMR | $0.674_{(0.056)}$ | $0.913_{(0.017)}$ | $0.914_{(0.004)}$ | $\underline{0.919}_{(0.012)}$ | $\mathbf{0.920}_{(0.013)}$ |
| $^{1}$H-NMR | $0.839_{(0.005)}$ | $0.935_{(0.031)}$ | $\underline{0.943}_{(0.036)}$ | $\mathbf{0.946}_{(0.027)}$ | $0.927_{(0.013)}$ |
| **IR** + ($^{13}$C-NMR, $^{1}$H-NMR) | $0.900_{(0.004)}$ | $0.936_{(0.013)}$ | $0.944_{(0.012)}$ | $\underline{0.947}_{(0.014)}$ | $\mathbf{0.959}_{(0.022)}$ |
| $^{13}$**C-NMR** + (IR, $^{1}$H-NMR) | $0.900_{(0.004)}$ | $0.936_{(0.013)}$ | $0.944_{(0.012)}$ | $\underline{0.947}_{(0.014)}$ | $\mathbf{0.957}_{(0.014)}$ |
| $^{1}$**H-NMR** + (IR, $^{13}$C-NMR) | $0.900_{(0.004)}$ | $0.936_{(0.013)}$ | $0.944_{(0.012)}$ | $\underline{0.947}_{(0.014)}$ | $\mathbf{0.956}_{(0.031)}$ |
| **IR** + (MS/MS$_{pos}$, MS/MS$_{neg}$) | $0.887_{(0.008)}$ | $0.911_{(0.003)}$ | $0.924_{(0.012)}$ | $\underline{0.931}_{(0.031)}$ | $\mathbf{0.944}_{(0.015)}$ |
| **SDBS Database (Experimental Spectra)** | | | | | |
| MS | $0.801_{(0.018)}$ | $0.826_{(0.021)}$ | $\underline{0.837}_{(0.015)}$ | $0.836_{(0.010)}$ | $\mathbf{0.847}_{(0.012)}$ |
| $^{13}$C-NMR | $0.729_{(0.033)}$ | $0.821_{(0.020)}$ | $\underline{0.833}_{(0.014)}$ | $0.836_{(0.011)}$ | $\mathbf{0.842}_{(0.015)}$ |
| $^{1}$H-NMR | $0.701_{(0.027)}$ | $0.779_{(0.025)}$ | $\underline{0.801}_{(0.019)}$ | $\mathbf{0.803}_{(0.018)}$ | $0.792_{(0.014)}$ |
| **MS** + ($^{13}$C-NMR, $^{1}$H-NMR) | $0.847_{(0.022)}$ | $0.858_{(0.019)}$ | $0.872_{(0.020)}$ | $\underline{0.881}_{(0.017)}$ | $\mathbf{0.913}_{(0.021)}$ |
| $^{13}$**C-NMR** + (MS, $^{1}$H-NMR) | $0.847_{(0.022)}$ | $0.858_{(0.019)}$ | $0.872_{(0.020)}$ | $\underline{0.881}_{(0.017)}$ | $\mathbf{0.894}_{(0.016)}$ |
| $^{1}$**H-NMR** + (MS, $^{13}$C-NMR) | $0.847_{(0.022)}$ | $0.858_{(0.019)}$ | $0.872_{(0.020)}$ | $\underline{0.881}_{(0.017)}$ | $\mathbf{0.909}_{(0.018)}$ |

MS/MS spectra. For 3D molecular conformation generation, we employ the QM9S dataset Zou et al. (2023), providing 130K molecules with UV, IR, and Raman spectra paired with their ground-truth 3D conformations. To examine model performance under experimental conditions, we collect about 12K molecules from the National Institute of Advanced Science and Technology, SDBS Web (https://sdbs.db.aist.go.jp) with MS, $^{13}$C-NMR, and 1H-NMR spectra, since no multi-modal dataset with experimental spectra is currently available. Further details are in Appendix G.

**Baselines.** For the molecular classification task, we compare our model against 1D-CNN Jung et al. (2023), Transformer Klein et al. (2018), models by Alberts et al. Alberts et al. (2025), Wu et al. Wu et al. (2025), SpectraLLM Shen et al. (2025), and DiffSpectra Wang et al. (2025). For conformation generation, as far as we known, no prior work has been published. We thus constructed two baselines. The first is to replace our spectral encoder with a standard attention module, while keeping the diffusion architecture and hyperparameters identical. The second adapts GeoDiff Xu et al. (2022), which generates five candidate conformations from SMILES, augmented with a contrastive selector to choose the one best matching the input spectrum.

**Metrics.** For the substructure classification task, we use micro F1-scores, molecular prediction accuracy, and Functional Group Similarity (FGSim). The latter is defined at the sample level. For the conformation generation task, performance is evaluated by the root mean square deviation (RMSD). All experiments are repeated 8 times with 8:1:1 dataset split, and the average results with variances are reported. We provide detailed hyperparameter settings in Appendix B and a full complexity analysis in Appendix H.

## 3.2 Model Performance on Structure Elucidation(RQ1)

We first evaluate the MSpecTmol's capability in chemical substructure prediction, which serves as a fundamental function of the framework. The detailed results are presented in Table 1 and Figure 3.

**Obs.1: MSpecTmol achieves superior performance in multi-modal settings.** MSpecTmol exhibits strong capability in identifying molecular functional groups and significantly outperforms baseline models, as shown in Table 1. In the single-modal setting, the performance of MSpecTmol is higher than transformer-based models, which demonstrates the effectiveness of our proposed archi-

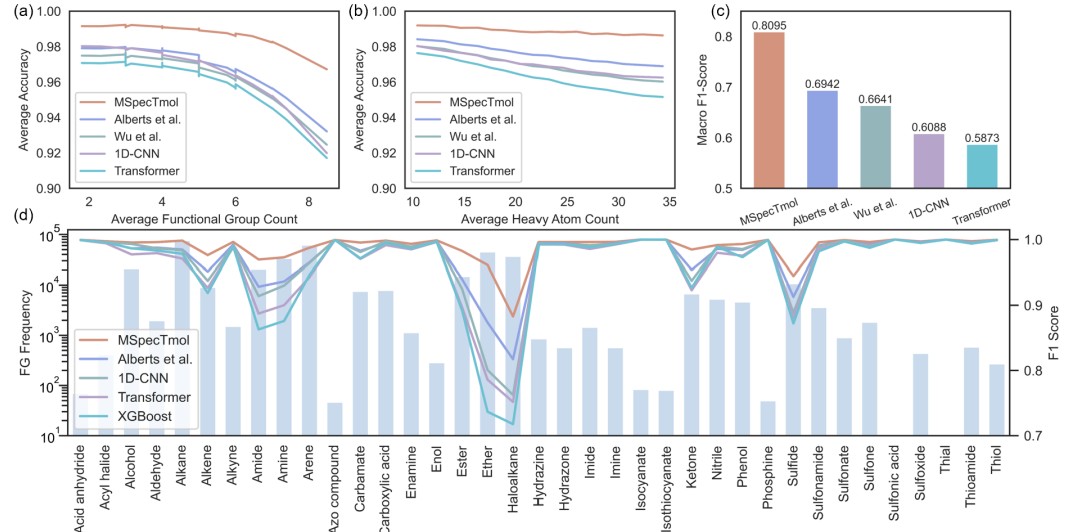

Figure 3: Comparison of our model against baselines in terms of accuracy and macro-F1 score. (a) Performance across different numbers of functional groups; (b) Performance under varying heavy atom counts; (c) Macro-F1 score results; (d) Correlation between the number of functional groups and prediction performance. Samples are binned in chunks of 5000.

tecture in capturing both local and global spectral features. However, when auxiliary spectra are introduced in the multi-modal setting, performance improves consistently across all models. Notably, MSpecTmol benefits the most, indicating its ability to effectively leverage additional information from auxiliary spectra to enhance substructure recognition and prediction accuracy. Furthermore, to demonstrate the broader applicability of MSpecTmol, we extended our evaluation to other state-of-the-art baselines with different objectives. Specifically, SpectraLLM leverages LLMs to treat spectral analysis as a sequence generation task, while DiffSpectra focuses on end-to-end molecular conformation generation. As shown in Table 2, MSpecTmol consistently outperforms both baselines in terms of Functional Group Similarity scores on the QM9S dataset, underscoring its robustness in identifying chemically meaningful substructures. In Appendix H, we analyze the computational overhead (time and memory) of MSpecTmol across varying numbers of input modalities and outline a decision framework for selecting the primary modality. To balance efficiency and performance, we selected an ideal combination of three modalities: IR, $^1$H-NMR, and $^{13}$C-NMR, which delivers strong predictive power with acceptable complexity.

Table 2: Functional Group Similarity comparison on the **QM9S dataset**. We compare MSpecTmol against generative baselines under their respective evaluation standards (17 functional groups for SpectraLLM, 13 for DiffSpectra).

| Modality | 17 Functional Groups | | 13 Functional Groups | |
|---|---|---|---|---|
| | SpectraLLM | MSpecTmol | DiffSpectra | MSpecTmol |
| IR | 0.6599 | **0.9328** | 0.9322 | **0.9501** |
| Raman | 0.7317 | **0.9334** | 0.9279 | **0.9417** |
| UV-Vis | 0.3713 | **0.5449** | 0.4354 | **0.5621** |
| All (IR+Ram.+UV) | 0.7934 | **0.9781** | 0.9495 | **0.9830** |

**Obs.2: MSpecTmol exhibits superior robustness under increasing structural complexity.** In the process of functional group classification, as shown in Figure 3(a) and (b), we observe that the prediction accuracy tends to decrease as the number of functional groups and heavy atoms in a molecule increases. This is likely because greater structural complexity leads to more intricate and overlapping spectral signals, making it challenging to disentangle the features corresponding to individual substructures. Despite these challenges, our model consistently achieves superior performance rel-

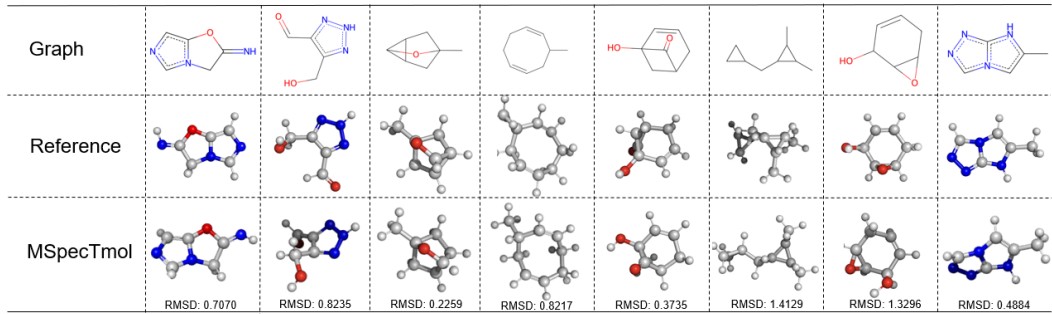

Figure 4: Example conformations generated from the QM9S dataset. The top row displays the 2D graphs, the middle row shows the reference structures, while the bottom shows the conformations generated by MSpecTmol with their corresponding RMSD values.

ative to baselines, particularly for molecules with complex structures. This advantage is primarily attributed to the PA-IB framework, which selectively filters out redundant or non-informative signals and retains only the most relevant structural information, thereby enhancing predictive reliability. To further validate the model's generalization capability on unseen complex structures, we performed a stress test on the top 10% largest molecules in the dataset. Results provided in Appendix K demonstrate that MSpecTmol maintains robust performance (F1-score of 0.925) even under significant distribution shifts.

**Obs.3: MSpecTmol exhibits superior performance across diverse functional groups.** As shown in Figure 3(c), MSpecTmol achieves the highest macro-F1 score, indicating balanced performance across both common and rare substructures. In classifying 37 functional groups, prediction accuracy varies due to intrinsic spectral differences. Groups with weak or overlapping signals, such as alkyl (-CH$_3$) and ether (-O-), are more challenging than those with distinct peaks like carbonyl (C=O) and hydroxyl (-OH). To provide more interpretable insights into these variations, we include a detailed analysis based on confusion matrices in Appendix P. Nevertheless, MSpecTmol consistently outperforms baselines, benefiting from the PA-IB framework that filters redundant signals and preserves the most informative structural cues.

**Obs.4: MSpecTmol exhibits superior performance on experimental spectra.** To evaluate the effectiveness of MSpecTmol on real-world data, we constructed a dataset of approximately 12K samples collected from the SDBS Web. As shown in Table 1, all models suffer from a noticeable performance drop under single-modality settings, which can be attributed to the limited amount of data and the inherent complexity of experimental spectra. However, in multi-modal settings, MSpecTmol achieves a substantial improvement over the baselines, reaching a F1 score of 0.913. This result highlights the practical value of MSpecTmol. Furthermore, given that CNNs-based model are inherently data-hungry and the scarcity of multi-modal datasets containing experimental spectra, we investigated various data augmentation strategies in Appendix Q, which yielded additional performance gains. Our analysis in Appendix R confirms that MSpecTmol is also highly robust against noisy and missing spectra, highlighting its reliability for practical deployment.

## 3.3 MODEL PERFORMANCE ON CONFORMATION GENERATION (RQ2)

In this section, we evaluate the effectiveness of MSpecTmol for spectrum-conditioned molecular conformation generation. Specifically, we integrated MSpecTmol as a spectral encoder into a diffusion model, which is trained to generate corresponding atom coordinates by input spectrum and and SMILES representations. The detailed algorithmic procedure is presented in the Appendix S.

**Obs.5: MSpecTmol facilitates high-fidelity conformation generation.** To thoroughly evaluate the benefit of incorporating spectral data, we compared MSpecTmol against both **spectrum-free graph-based methods** (RDKit, OpenBabel, ConfGF) and **spectral-conditioned baselines** (GeoDiff, Attention-based model).As shown in Table 3, MSpecTmol consistently outperforms both baselines across all spectral inputs. This performance advantage becomes more pronounced when multi-

Table 3: Comparison of mean RMSD for molecular conformation generation. We evaluate baselines including graph-based methods (RDKit, OpenBabel, ConfGF) and spectral-conditioned models. Lower RMSD indicates better agreement with reference structures.

| Input Spectra | Graph-based Baselines | | | Spectral-Conditioned Models | | |
|---|---|---|---|---|---|---|
| | RDKit | OpenBabel | ConfGF | GeoDiff | Attention | MSpecTmol |
| UV | 1.350 | 1.279 | 1.143 | $1.125_{(0.022)}$ | $0.718_{(0.009)}$ | $0.697_{(0.007)}$ |
| IR | 1.350 | 1.279 | 1.143 | $1.233_{(0.020)}$ | $0.726_{(0.010)}$ | $0.706_{(0.009)}$ |
| Raman | 1.350 | 1.279 | 1.143 | $1.042_{(0.032)}$ | $0.735_{(0.011)}$ | $0.701_{(0.008)}$ |
| UV + (IR, Raman) | 1.350 | 1.279 | 1.143 | $0.882_{(0.019)}$ | $0.714_{(0.008)}$ | $0.682_{(0.006)}$ |
| IR + (UV, Raman) | 1.350 | 1.279 | 1.143 | $0.882_{(0.019)}$ | $0.714_{(0.007)}$ | $0.689_{(0.007)}$ |

modal spectra are utilized, indicating that our model generates conformations that are highly consistent with the input spectral data. This advantage stems from MSpecTmol's superior ability to fuse information from multiple spectra into a rich, unified representation for the generative task. Furthermore, the qualitative results in Figure 4 corroborate these findings, illustrating our model's capacity to translate complex spectral patterns into high-fidelity molecular structures.

**Obs.6: MSpecTmol achieves superior stability through effective multi-modal fusion.** Figure 6 demonstrates the superior performance of MSpecTmol, which consistently generates conformations with a lower and more tightly concentrated RMSD distribution than all baselines. This result highlights the necessity of multi-modal fusion, as single-modality models yield higher errors and greater variance. Moreover, the stark performance degradation observed in the ablation variant, which is evidenced by its high and widely dispersed RMSD—confirms that our proposed fusion architecture is critical for ensuring both the accuracy and stability of the generative process.

### 3.4 INTERPRETALITY AND ABLATION STUDY AND SENSITIVITY ANALYSIS (RQ3)

In this section, we further investigate the intrinsic relationships between different spectral segments and molecular substructures. Additionally, we analyze the contributions of individual model components and examine the model's sensitivity to hyperparameter variations, detailed in Figure 11.

**Obs.7: Different spectral modalities emphasize distinct molecular features.** To probe the relationship between spectral segments and molecular substructures, we designed an experiment where a model predicts a single functional group from a concatenated input of three spectra. By averaging the importance scores in the dataset, we obtained attention distribution images for the three modalities, as shown in Figure 5. In the overall image, we can observe that different functional groups tend to focus on different spectra, which may indicate that each spectrum contains information with a distinct emphasis. The varying roles of spectral information highlight the necessity of utilizing information bottleneck theory to extract supplementary information from auxiliary modalities that enhance the main modality. More results could be found in Appendix O. Additionally, to demonstrate the superiority of our PA-IB framework over conventional integration paradigms, we compared MSpecTmol against various standard fusion strategies. Detailed comparisons and analysis are provided in Appendix N.

**Obs.8: $\alpha$ and $\beta$ regulate the trade-off between prediction accuracy and information compression.** We analyze the joint effect of $\alpha$ and $\beta$ in balancing prediction accuracy and information compression, as defined in equation 1. As shown in Figure 11(a), setting $\alpha = \beta = 1 \times 10^{-6}$ consistently yields the best performance. Larger values of $\alpha$ and $\beta$ impose excessive compression, hindering the model's ability to retain crucial essential spectral features. Conversely, smaller values preserve more input information but fail to effectively suppress redundancy, ultimately compromising generalization. These results highlight the critical importance of properly tuning $\alpha$ and $\beta$ to maintain an optimal trade-off.

**Obs.9: Regulating Information Processing at Different Levels via the Primary–Auxiliary Information Bottleneck.** We conducted ablation studies on the multi-modal spectroscopic dataset. As shown in Figure 11(b), with all loss terms, the model achieved an F1-score of 0.9589. Removing the KL divergence loss for primary spectra ($\mathcal{L}_{MI^2}$), which regulates compression in the primary modal-

Figure 5: Illustration of spectral information importance. (a) Molecular structure of `CC#COc1ccc(CCC(=O)Cl)cc1OC`, which contains Alkane, Ether, and Haloalkane functional groups. (b) Importance map of IR spectra, (c) $^{13}$C-NMR spectra, (d) and $^{1}$H-NMR spectra. The x-axis of all three spectral plots is normalized to the range [0, 150].

Table 4: Performance comparison (F1-score) between our asymmetric PA-IB and the symmetric Uniform IB baseline across different modality configurations.

| Fusion Strategy | IR | $^{13}$C-NMR | $^{1}$H-NMR | Multi-Modal (All) |
|---|---|---|---|---|
| Symmetric (Uniform IB) | 0.906 | 0.904 | 0.905 | 0.934 |
| **Asymmetric (Ours)** | **0.923** | **0.920** | **0.927** | **0.959** |

ity, reduced the F1-score to 0.9543. Removing the KL divergence for auxiliary spectra ($\mathcal{L}_{MI^1}$ and $\mathcal{L}_{MI^3}$) led to a more substantial decrease to 0.9521, indicating that uncompressed auxiliary information introduces noise that undermines prediction quality. These results underscore the importance of regulating information at both levels to suppress redundancy and preserve relevant features.

**Obs.10: Superiority of Asymmetric Design over Symmetric Fusion.** To demonstrate the effectiveness of our asymmetric fusion strategy, we compared MSpecTmol against a symmetric Uniform IB baseline, which applies uniform information bottleneck constraints across all modalities independently without the primary-auxiliary distinction. As presented in Table 4, MSpecTmol consistently outperforms the symmetric approach. Notably, in the full multi-modal setting, our method achieves a 2.5% absolute gain (0.959 vs. 0.934), confirming that the asymmetric design effectively suppresses cross-modal redundancy that the symmetric strategy fails to address.

## 4 CONCLUSION AND FUTURE OUTLOOK

In this work, we introduce **MSpecTmol**, a multi-modal spectrum information fusion framework based on the information bottleneck principle, designed for molecular structure determination. Our framework adopts a primary-auxiliary synergistic modeling approach, which distills core information from a primary modality while leveraging auxiliary spectra to supplement and refine the final representation. Rigorous experimental evaluations validate MSpecTmol's end-to-end effectiveness, achieving a SOTA F1-score of 0.959 in molecular identification and a low average RMSD of 0.682Å in 3D conformation generation. Meanwhile, our model provides chemically interpretable spectroscopic fragment importance, bridging the gap between ML predictions and domain knowledge.

Looking forward, this framework not only assists chemists in unraveling complex molecular systems but also accelerates the analysis of novel compounds. MSpecTmol holds potential to benefit diverse scientific domains—such as drug discovery, materials science, and chemical forensics—where accurate and reliable molecular identification is critical. MSpecTmol paves the way toward democratized, efficient, and interpretable molecular analysis for broad scientific and industrial applications.

## 5 REPRODUCIBILITY

We provide the complete implementation in the repository along with guidance on how to reproduce our results. Our code is available at `https://anonymous.4open.science/r/MspecTmol-6B4D`.

## 6 ETHICS STATEMENT

Our study does not involve human participants, personal data, or sensitive information. The datasets and resources used are either publicly available or released under appropriate licenses. We confirm that our research does not raise any ethical concerns related to privacy, safety, fairness, or potential misuse. The contributions of this work are intended solely for advancing scientific research and are not designed or evaluated for harmful applications.

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

## A    Use of Large Language Models (LLMs)

In preparing this manuscript, we used a large language model (LLM) solely for writing assistance and text refinement (e.g., grammar correction, stylistic polishing, and conciseness). The LLM was not involved in research design, data analysis or model implementation. All technical content, experiments, and conclusions were conceived, executed, and validated by the authors.

## B    Training Settings

For the functional group classification task, the model was trained for 100 epochs with a batch size of 128, using the Adam optimizer with an initial learning rate of $4 \times 10^{-4}$ and a cosine annealing scheduler. The loss coefficients $\alpha$ and $\beta$ were set to $1 \times 10^{-6}$, while the weight for the auxiliary prediction loss was 0.7. The temperature for the information bottleneck's stochastic gating was maintained at 1.0. For the spectrum-conditioned molecular conformation generation task, the model was trained for 10,000 iterations with a batch size of 64. We employed the Adam optimizer with a learning rate of $1 \times 10^{-3}$, which was adjusted by a plateau scheduler based on validation loss. For this task, the loss coefficients $\alpha$ and $\beta$ were both set to $1 \times 10^{-6}$. Both models were trained on two NVIDIA A100 GPUs (80 GB each). The classification model required approximately 6 hours of training, while the conformation generation model took around 30 hours.

## C    Related work

## D    Related work

### D.1    Spectroscopy-Based Molecular Modeling

Machine learning has advanced spectroscopy-based molecular structure prediction significantly. Liu et al. introduced MS2SMILES Liu et al. (2023), treating hydrogen atoms as implicitly linked to heavy atoms, improving molecular generation accuracy. Ji et al. Ji et al. (2020) presented DeepEI, a deep learning framework for elucidating structures from EI-MS spectra. Wei et al Wei et al. (2019). developed NEIMS, a neural network model that captures fragmentation patterns from electron ionization for rapid small molecule mass spectrum prediction. Marcus proposed ZODIAC Ludwig et al. (2020), leveraging tandem mass spectrometry (MS/MS) for molecular formula generation. Michael A. et al. Stravs et al. (2022) further explored de novo molecular structure generation using RNN models.

More recently, the rapid development of generative AI has expanded spectral analysis into the realm of Large Language Models and Diffusion Models. For instance, DiffSpectra Wang et al. (2025) introduces a generative framework that formulates structure elucidation as a conditional diffusion process, enabling the end-to-end generation of 3D molecular conformations from multi-modal spectra. In parallel, Large Language Models have been adapted for this domain: SpectraLLM Su et al. (2025) and MolSpectLLM Shen et al. (2025) leverage the reasoning capabilities of heavy transformer backbones to treat spectrum-to-structure translation as a sequence generation task, bridging spectroscopy with textual molecular representations. Furthermore, SpectrumWorld Yang et al. (2025) expands this frontier by introducing a multi-modal agent framework and benchmark suite to systematize deep learning research in spectroscopy.

While these methods have achieved notable success, previous approaches often rely heavily on single modalities like mass spectrometry or require massive computational resources. Additionally, mass spectrometry is costly, sensitive to interference, and challenging to standardize in automated workflows. Recently, Marvin et al. Alberts et al. (2024) released a 790k Multimodal Spectroscopic Dataset, providing a foundation for integrating multi-spectroscopic data. Their work introduced baseline models for tasks like molecular structure prediction and functional group identification from spectral data, forming a key resource for our research. These studies highlight the potential and limitations of current methods, motivating our approach to integrate multi-spectroscopic modalities for enhanced molecular structure determination.

## D.2 Information Bottleneck (IB) Theory

The IB theory provides a principled framework for extracting compact and informative substructures from complex data, playing a key role in challenges like denoising and compression. PGIB Yu et al. (2020) extends IB by introducing a framework with a mutual information estimator for irregular graph data, and a connectivity loss to stabilize information extraction. VGIB Yu et al. (2022) further improves stability by injecting Gaussian noise into node representations, regulating information flow between original and perturbed graphs. Lee et al. Lee et al. (2023) expanded IB to paired graphs with the Conditional GIB, optimizing compressed information extraction by retaining only the most relevant information. While effective, these approaches focus on single-target tasks and lack strategies for redundancy reduction and complementary integration under multi-modal conditions. This growing body of work underscores the versatility of IB theory while highlighting opportunities for further refinement, particularly in handling multi-modal scenarios, where redundancy removal and cross-modal synergy are essential.

# E Broader Impacts and Limitation Discussion

## E.1 Broader Impacts

This work promotes automated and interpretable molecular structure elucidation via multi-modal spectroscopic learning. MSpecTmol may assist domains such as drug discovery, materials science, and chemical forensics by providing chemically intuitive insights and reducing reliance on manual spectral interpretation. Its interpretable design supports broader and more accessible molecular analysis. The proposed framework can reduce reliance on extensive manual spectral interpretation, democratizing molecular analysis for broader scientific and industrial use.

## E.2 Limitations

While MSpecTmol demonstrates strong performance, several limitations remain. First, the model's effectiveness depends on the availability of complete, multi-modal spectra, which are often scarce in practice and may hinder its deployment on incomplete datasets. Additionally, its training on a fixed vocabulary of functional groups restricts its ability to identify rare or novel substructures, particularly when analyzing new chemical entities. Future work will focus on addressing these challenges to improve the model's robustness and expand its chemical scope.

# F Proof

## F.1 Proof of Proposition

$p_\theta(\mathbf{Y}|T_m)$ is variational approximation of $p(\mathbf{Y}|T_m)$. We model $p_\theta(\mathbf{Y}|T_m)$ as a predictor parametrized by $\theta$, which outputs the model prediction $\mathbf{Y}$ based on the core primary spectra $T_m$.

$$
\begin{aligned}
I(\mathbf{Y}; T_m) &= \mathbb{E}_{\mathbf{Y}, T_m}[\log \frac{p(\mathbf{Y}|T_m)}{p(\mathbf{Y})}] \\
&= \mathbb{E}_{\mathbf{Y}, T_m}[\log \frac{p_\theta(\mathbf{Y}|T_m)}{p(\mathbf{Y})}] \\
&\quad + \mathbb{E}_{T_m}[KL(p(\mathbf{Y}|T_m)||p_\theta(\mathbf{Y}|T_m))]
\end{aligned}
\tag{15}
$$

According to the non-negativity of the KL divergence, we have:

$$
\begin{aligned}
I(\mathbf{Y}; T_m) &\geq \mathbb{E}_{\mathbf{Y}, T_m}[\log \frac{p_\theta(\mathbf{Y}|T_m)}{p(\mathbf{Y})}] \\
&= \mathbb{E}_{\mathbf{Y}, T_m}[\log p_\theta(\mathbf{Y}|T_m)] + H(\mathbf{Y})
\end{aligned}
\tag{16}
$$

Thus, we can minimize the upper bound of $-I(\mathbf{Y}; T_m)$ by minimizing the model prediction loss $\mathcal{L}_{\text{pred}}(\mathbf{Y}, T_m)$, which can be modeled as the cross entropy loss for classification and the mean square loss for regression.

### F.1.1 MINIMIZING $-I(Y; T_m)$

(Upper bound of $-I(Y; T_m)$) Given the primary spectra $X_m$, and its label information $\mathbf{Y}$, we have

$$
\begin{aligned}
- I(\mathbf{Y}; T_m) &\leq \mathbb{E}_{T_m, \mathbf{Y}}[-\log p_\theta(\mathbf{Y}|T_m)] \\
&= \mathbb{E}_{(\mathbf{Y}; T_m)} \log \left[ P_\theta\left(\mathbf{Y} \mid T_m\right)\right] + H(\mathbf{Y}) := \mathcal{L}_{pred},
\end{aligned}
\tag{17}
$$

where $H(\mathbf{Y})$ is constant across all data, it will be omitted in the model optimization process. $p_\theta(\mathbf{Y}|T_m)$ is variational approximation of $p(\mathbf{Y}|T_m)$. A detailed proof for proposition is given in Appendix F.1.2.

### F.1.2 MINIMIZING $-I(Y; T_a \mid T_m)$

For the second term of Equation 9, i.e., $-I(Y; T_a \mid T_m)$, we decompose the term into the sum of two terms based on the chain rule of mutual information as follows:

$$
I(Y; T_a|T_m) = I(Y; T_a, T_m) - I(T_a; T_m).
\tag{18}
$$

For the upper bound of $-I(Y; T_a, T_m)$, Given the core primary spectra $T_m$ and core auxiliary spectra $T_a$, and its label information $\mathbf{Y}$, we have

$$
\begin{aligned}
- I(\mathbf{Y}; T_a, T_m) &\leq \mathbb{E}_{(T_a, T_m, \mathbf{Y})}[-\log p_\theta(\mathbf{Y}|T_a, T_m)] \\
&= \mathbb{E}_{(\mathbf{Y}, T_a, T_m)} \log \left[ P_\theta\left(\mathbf{Y} \mid T_a, T_m\right)\right] + H(\mathbf{Y}) := \mathcal{L}_{sup},
\end{aligned}
\tag{19}
$$

where $p_\theta(\mathbf{Y}|T_a, T_m)$ is variational approximation of $p(\mathbf{Y}|T_a, T_m)$. We model $p_\theta(\mathbf{Y}|T_a, T_m)$ as a predictor parametrized by $\theta$, which outputs the model prediction $\mathbf{Y}$ based on the core spectra $T_a$ and $T_m$. Thus, we can minimize the upper bound of $-I(\mathbf{Y}; T_a, T_m)$ by minimizing the supplementary prediction loss $\mathcal{L}_{\sup}$,

For the upper bound of $I(T_a; T_m)$, drawing inspiration from the experiences derived in Variational Autoencoders (VAE) Kingma (2013), we attempt to replace $p(t_a)$ with $q(t_a)$ and consolidate the additional components to form a Kullback-Leibler (KL) divergence:

$$
\begin{aligned}
I(T_a; T_m) &= \mathbb{E}_{(t_m, t_a) \sim p(t_m, t_a)} \left[ \log \frac{p(t_a \mid t_m)}{p(t_a)} \right] \\
&= \mathbb{E}_{(t_m, t_a) \sim p(t_m, t_a)} \left[ \log \frac{p(t_a \mid t_m)}{q(t_a)} \cdot \frac{q(t_a)}{p(t_a)} \right] \\
&= \mathbb{E}_{(t_m, t_a) \sim p(t_m, t_a)} \left[ \log \frac{p(t_a \mid t_m)}{q_{(t_a)}} \right] \\
&\quad + \mathbb{E}_{(t_m, t_a) \sim p(t_m, t_a)} \left[ \log \frac{q(t_a)}{p(t_a)} \right]
\end{aligned}
\tag{20}
$$

For the first term, both $p(t_a \mid t_m)$ and $q(t_a)$ have analytical forms, allowing the function within the brackets to be computed analytically. By utilizing the relationship $p(t_m, t_a) = p(t_m)p(t_a \mid t_m)$, we can rewrite the first term in a more elegant manner:

$$
\begin{aligned}
\mathbb{E}_{(t_m, t_a) \sim p(t_m, t_a)} &\left[ \log \frac{p(t_a \mid t_m)}{q(t_a)} \right] \\
&= \iint p(t_m)p(t_a \mid t_m) \log \frac{p(t_a \mid t_m)}{q(t_a)} \, dt_a \, dt_m \\
&= \int p(t_m) \left( \int p(t_a \mid t_m) \log \frac{p(t_a \mid t_m)}{q(t_a)} \, dt_a \right) dt_m \\
&= \mathbb{E}_{t_m \sim p(t_m)} \left[ \mathrm{KL}\left(p(t_a \mid t_m) \| q(t_a)\right) \right] \\
&:= L_{MI^1} \approx \frac{1}{N} \sum_{i=1}^{N} \mathrm{KL}\left[ p(t_a \mid t_{mi}) \| q(t_a) \right], \quad t_{mi} \sim p(t_m)
\end{aligned}
\tag{21}
$$

The term $L_{MI1} := \mathbb{E}_{t_m \sim p(t_m)}[\mathrm{KL}(p(t_a \mid t_m) \| q(t_a))]$ is often referred to as the *rate* in rate-distortion theory. This rate component can be optimized using mini-batch gradient descent. Specifically, by sampling a batch of training samples $t_{m1}, \ldots, t_{mN}$ from the training set, we can minimize the KL divergence $\mathrm{KL}\left[p(t_a \mid t_{mi}) \| q(t_a)\right]$ for each $t_{mi}$.

Since both distributions $p(t_a \mid t_m)$ and $q(t_a)$ are Gaussian, the KL divergence between them has an analytical solution:

$$
\begin{aligned}
&\mathrm{KL}\left[p(t_a \mid t_m)\|q(t_a)\right] \\
&= \mathrm{KL}\left[\mathcal{N}\left(\boldsymbol{\mu}(t_m), \boldsymbol{\sigma}^2(t_m)\boldsymbol{I}\right)\|\mathcal{N}(\mathbf{0}, \boldsymbol{I})\right] \\
&= \sum_{j=1}^{J} \mathrm{KL}\left[\mathcal{N}\left(\mu_j, \sigma_j^2\right)\|\mathcal{N}(0, 1)\right] \\
&= \sum_{j=1}^{J} \frac{1}{2}\left(-\log\sigma_j^2 - 1 + \mu_j^2 + \sigma_j^2\right)
\end{aligned}
\tag{22}
$$

Here, $\mu(t_m), \sigma^2(t_m)$ are mean and variance of $\mathbf{H}^m$, respectively.

### F.1.3 MINIMIZING $- I\left(X_m; T_m\right)$

For the upper bound of $- I\left(X_m; T_m\right)$, Similarly, we attempt to replace $p(t_m)$ with $q(t_m)$ and consolidate the additional components to form a Kullback-Leibler (KL) divergence:

$$
\begin{aligned}
I(X_m; T_m) &= \mathbb{E}_{(t_m, x_m)\sim p(t_m, x_m)}\left[\log\frac{p(t_m \mid x_m)}{q_{(t_m)}}\right] \\
&\quad + \mathbb{E}_{(t_m, x_m)\sim p(t_m, x_m)}\left[\log\frac{q(t_m)}{p(t_m)}\right]
\end{aligned}
\tag{23}
$$

By utilizing the relationship $p(t_m, x_m) = p(x_m)p(t_m \mid x_m)$, we can rewrite the first term in a more elegant manner:

$$
\begin{aligned}
&\mathbb{E}_{(t_m, x_m)\sim p(t_m, x_m)}\left[\log\frac{p(t_m \mid x_m)}{q(t_m)}\right] \\
&= \mathbb{E}_{t_m\sim p(t_m)}\left[\mathrm{KL}\left(p(t_m \mid x_m)\|q(t_m)\right)\right] := L_{MI^2}
\end{aligned}
\tag{24}
$$

The term $L_{MI^2} := \mathbb{E}_{t_m\sim p(t_m)}[\mathrm{KL}(p(t_m \mid x_m)\|q(t_m))]$ is often referred to as the *rate* in rate-distortion theory. This rate component can be optimized using mini-batch gradient descent. Specifically, by sampling a batch of training samples $t_{m1}, \ldots, t_{mN}$ from the training set, we can minimize the KL divergence $\mathrm{KL}\left[p(t_m \mid x_{mi})\|q(t_m)\right]$ for each $x_{mi}$.

Since both distributions $p(t_m \mid x_m)$ and $q(t_m)$ are Gaussian, the KL divergence between them has an analytical solution:

$$
\mathrm{KL}\left[p(t_m \mid x_m)\|q(t_m)\right] = \sum_{j=1}^{J} \frac{1}{2}\left(-\log\sigma_j^2 - 1 + \mu_j^2 + \sigma_j^2\right)
\tag{25}
$$

### F.1.4 MINIMIZING $- I\left(T_a; X_m, X_a\right)$

For the upper bound of $- I\left(T_a; X_m, X_a\right)$, Similarly, we attempt to replace $p(t_a)$ with $q(t_a)$ and consolidate the additional components to form a Kullback-Leibler (KL) divergence:

$$
\begin{aligned}
I(T_a; X_m, X_a) &= \mathbb{E}_{(t_a, x_a, x_m)\sim p(t_a, x_a, x_m)}\left[\log\frac{p(t_a \mid x_m, x_a)}{q_{(t_a)}}\right] \\
&\quad + \mathbb{E}_{(t_a, x_a, x_m)\sim p(t_a, x_a, x_m)}\left[\log\frac{q(t_a)}{p(t_a)}\right]
\end{aligned}
\tag{26}
$$

By utilizing the relationship $p(t_a, x_a, x_m) = p(t_a, x_a)p(t_a \mid x_m, x_a)$, we can rewrite the first term in a more elegant way:

$$
\begin{aligned}
&\mathbb{E}_{(t_a, x_a, x_m)\sim p(t_a, x_a, x_m)}\left[\log\frac{p(t_a \mid x_m, x_a)}{q(t_a)}\right] \\
&= \mathbb{E}_{t_a, x_a\sim p(x_m, x_a)}\left[\mathrm{KL}\left(p(t_a \mid x_m, x_a)\|q(t_a)\right)\right] := L_{MI^3}
\end{aligned}
\tag{27}
$$

The term $L_{MI3} = \mathbb{E}_{x_m, x_a\sim p(x_m, x_a)}[\mathrm{KL}(p(t_a \mid x_m, x_a)\|q(t_a))]$ is often referred to as the *rate* in rate-distortion theory. This rate component can be optimized using mini-batch gradient descent.

Specifically, by sampling a batch of training samples $t_{m1}, \ldots, t_{mN}$ from the training set, we can minimize the KL divergence $\text{KL}\left[p\left(t_a \mid t_{mi}\right) \| q(t_a)\right]$ for each $t_{mi}$.

Since both distributions $p(t_a \mid x_m, x_a)$ and $q(t_a)$ are Gaussian, the KL divergence between them has an analytical solution:

$$\text{KL}\left[p(t_a \mid x_m, x_a) \| q(t_a)\right] = \sum_{j=1}^{J} \frac{1}{2}\left(-\log \sigma_j^2 - 1 + \mu_j^2 + \sigma_j^2\right) \tag{28}$$

## G  DEFINITION OF FUNCTIONAL GROUPS

Functional groups play a crucial role in determining the chemical reactivity and properties of molecules. To systematically analyze molecular structures, we employ a set of predefined patterns to identify key functional groups within a given molecular dataset.

Table 5 lists the functional groups considered in this study, along with their corresponding SMARTS representations. These functional groups were selected based on their relevance to organic and medicinal chemistry, including common moieties such as hydroxyl (-OH), carbonyl (C=O), and amine (-NH$_2$) groups. The identification process involves scanning molecular structures using sub-graph matching algorithms, ensuring accurate detection of these structural motifs.

Table 5: Predefined Functional Groups and Their SMARTS Patterns

| Functional Group | SMARTS Pattern |
| --- | --- |
| Acid anhydride | `[CX3](=[OX1])[OX2][CX3](=[OX1])` |
| Acyl halide | `[CX3](=[OX1])[F,Cl,Br,I]` |
| Alcohol | `[#6][OX2H]` |
| Aldehyde | `[CX3H1](=O)[#6,H]` |
| Alkane | `[CX4;H3,H2]` |
| Alkene | `[CX3]=[CX3]` |
| Alkyne | `[CX2]#[CX2]` |
| Amide | `[NX3][CX3](=[OX1])[#6]` |
| Amine | `[NX3;H2,H1,H0;!$(NC=O)]` |
| Arene | `[cX3]1[cX3][cX3][cX3][cX3][cX3]1` |
| Azo compound | `[#6][NX2]=[NX2][#6]` |
| Carbamate | `[NX3][CX3](=[OX1])[OX2H0]` |
| Carboxylic acid | `[CX3](=O)[OX2H]` |
| Enamine | `[NX3][CX3]=[CX3]` |
| Enol | `[OX2H][#6X3]=[#6]` |
| Ester | `[#6][CX3](=O)[OX2H0][#6]` |
| Ether | `[OD2]([#6])[#6]` |
| Haloalkane | `[#6][F,Cl,Br,I]` |
| Hydrazine | `[NX3][NX3]` |
| Hydrazone | `[NX3][NX2]=[#6]` |
| Imide | `[CX3](=[OX1])[NX3][CX3](=[OX1])` |
| Imine | `[$([CX3]([#6])[#6]),$([CX3H][#6])]=[$([NX2][#6]),$([NX2H])]` |
| Isocyanate | `[NX2]=[C]=[O]` |
| Isothiocyanate | `[NX2]=[C]=[S]` |
| Ketone | `[#6][CX3](=O)[#6]` |
| Nitrile | `[NX1]#[CX2]` |
| Phenol | `[OX2H][cX3]:[c]` |
| Phosphine | `[PX3]` |
| Sulfide | `[#16X2H0]` |
| Sulfonamide | `[#16X4]([NX3])(=[OX1])(=[OX1])[#6]` |
| Sulfonate | `[#16X4](=[OX1])(=[OX1])([#6])[OX2H0]` |
| Sulfone | `[#16X4](=[OX1])(=[OX1])([#6])[#6]` |
| Sulfonic acid | `[#16X4](=[OX1])(=[OX1])([#6])[OX2H]` |
| Sulfoxide | `[#16X3]=[OX1]` |
| Thial | `[CX3H1](=S)[#6,H]` |
| Thioamide | `[NX3][CX3]=[SX1]` |
| Thiol | `[#16X2H]` |

The functional group identification is performed using cheminformatics libraries such as RDKit, which allows for efficient substructure searches within molecular datasets. This approach enables us to extract chemically meaningful information and facilitate downstream tasks such as molecular property prediction, reactivity analysis, and structure-based clustering.

## H    COMPLEXITY ANALYSIS AND CHOICE OF PRIMARY SPECTRA

We conducted a comprehensive analysis to determine the optimal multi-modal configuration for MSpecTmol, balancing predictive performance with computational efficiency. The time and space complexity of our model and several baselines are presented in Table 6. This analysis reveals a clear trade-off between the number of input modalities and the required resources. As shown in table 7,while expanding from three to five spectral inputs nearly doubled the resource consumption, it yielded only marginal performance improvements. This finding led us to select a three-modality fusion as the most balanced and efficient configuration.

A critical aspect of our framework is the selection of the primary spectrum, as MSpecTmol is designed to prioritize its features while using auxiliary spectra for supplementary information. Our recommended procedure is to first identify the single best-performing modality in standalone experiments and assign it the primary role, thereby ensuring that the most informative stream is preserved.

To implement this strategy for the functional group classification task, we first assessed the predictive power of each individual spectrum (Table 1 and Figure 9(a)). The results revealed that IR spectroscopy delivered relatively high and stable accuracy, making it the ideal candidate for the primary spectrum. Conversely, MS/MS spectra exhibited the lowest performance. In our multi-modal evaluations, we observed that fusing multiple spectra consistently improved performance. Notably, the combination of IR, $^1$H-NMR, and $^{13}$C-NMR not only outperformed other fusion strategies—achieving the best results for 35 out of 37 functional groups—but was also more effective than using all available spectra, all while maintaining lower computational complexity (Figure 9(b)).

Consequently, we established the optimal configuration for this task as using IR spectroscopy as the primary input, with the $^1$H-NMR and $^{13}$C-NMR modalities serving as powerful auxiliary inputs.

Table 6: Comparison of resource usage and performance.

| Model | Mem.(GB) | Time(h) | F1-score |
|---|---|---|---|
| 1D-CNN | 5.7 | 2 | 0.900 |
| Transformer | 1.7 | 35 | 0.911 |
| MSpecTmol | 6 | 3 | 0.959 |

Table 7: Comparison across different numbers of modalities.

| #Mod. | Mem.(GB) | Time(h) | F1-score |
|---|---|---|---|
| 1 | 3.4 | 2 | 0.923 |
| 3 | 6.6 | 2.5 | 0.959 |
| 5 | 11.3 | 5 | 0.963 |

Table 8: Inference time for processing 10,000 samples.

| Model | Inference Time (s) |
|---|---|
| 1D-CNN | 10.4 |
| Transformer | 45.1 |
| Wu et al. | 55.4 |
| Alberts et al. | 60.1 |
| **MSpecTmol** | **14.0** |

Table 9: Training time vs. Molecular Size (Heavy Atom Count).

| Heavy Atom Count | Training Time (s) |
|---|---|
| 5 - 15 | 564 |
| 16 - 25 | 556 |
| 25 - 35 | 558 |

## I    INFERENCE TIME AND SCALABILITY

**Inference Time**    To assess the model's suitability for high-throughput screening, we measured the total inference time for processing 10,000 samples on a single NVIDIA A100 GPU. Table 8 shows that MSpecTmol completes the task in just 14.0 seconds. This speed is comparable to the simple 1D-CNN (10.4 s) and drastically faster than the Transformer (45.1s), confirming its efficiency for real-time applications.

**Scalability with Molecular Size**   To verify whether the model's computational cost is sensitive to molecular complexity, we measured the training time on subsets of data sorted by Heavy Atom Coun. As presented in Table 9, the training time remains remarkably consistent ($\approx$ 560s) across different molecular sizes. This is because our model takes fixed-dimension interpolated spectra as input and outputs functional group probabilities; consequently, the physical size or complexity of the molecule does not alter the input tensor dimensions or the model architecture.

## J  SENSITIVITY ANALYSIS OF GUMBEL-SOFTMAX TEMPERATURE

In the Core Spectrum Extraction module, we employ Gumbel-Softmax to enable differentiable sampling of the discrete importance masks. The temperature parameter $t$ plays a pivotal role in controlling the sharpness of this distribution. To evaluate its impact on model performance, we conducted a sensitivity analysis across various temperatures. As shown in Table 10, MSpecTmol achieves the optimal F1-score at $t = 1.0$. When the temperature is set to a lower value ($t = 0.5$), the performance experiences a slight decline. Conversely, increasing the temperature to higher values ($t = 1.5, 2.0$) leads to a more noticeable degradation in prediction quality.

Table 10: Sensitivity analysis of the Gumbel-Softmax temperature parameter $t$.

| Temperature ($t$) | F1-score |
|---|---|
| 0.5 | 0.952 |
| **1.0** | **0.959** |
| 1.5 | 0.954 |
| 2.0 | 0.945 |

Intuitively, the temperature controls the sparsity and sharpness of the gating over spectral frequency bands. In our framework, each gate determines whether a local region of the spectrum is preserved or replaced by noise. A very low temperature makes these decisions almost binary. While this promotes sparsity, it risks discarding weak but chemically informative peaks (e.g., small shoulders or minor bands) that are critical for distinguishing fine-grained functional groups and isomers. On the other hand, a high temperature yields overly soft gates, causing most bands to be partially retained. This weakens the model's ability to suppress redundancy and blurs the importance patterns across modalities. The superior performance observed at $t = 1.0$ confirms that this setting achieves an optimal balance, allowing the PA-IB framework to learn selective yet stable masks that retain structurally informative spectral regions while effectively filtering out redundant or noisy segments.

## K  GENERALIZATION ANALYSIS ON COMPLEX MOLECULAR STRUCTURES

To evaluate the model's generalization capability on samples with more complex molecular structures and denser, overlapping spectral peaks, we performed a supplementary stress test on the dataset from Alberts et al. Specifically, instead of a standard random split, we sorted the entire dataset by heavy atom count. We utilized the bottom 90% for training and reserved the top 10% strictly for testing. This setup introduces a significant distribution shift, requiring the model to infer the structure of complex molecules that are physically larger than any sample seen during training.

The results are presented in Table 11. While the performance on unseen larger molecules naturally dips compared to the standard random split due to the increased structural complexity, MSpecT-mol maintains a high F1-score of 0.925. Notably, our model consistently outperforms baselines in this challenging setting. This demonstrates that our PA-IB framework effectively learns intrinsic spectroscopic-structural correlations rather than simply memorizing dataset-specific patterns, confirming its capability to generalize to more complex chemical spaces.

## L  ANALYSIS OF PRIOR DISTRIBUTION CHOICE FOR LATENT VARIABLES

In our Primary-Auxiliary Information Bottleneck (PA-IB) framework, the choice of the prior distributions for the latent bottleneck variables $T_m$ and $T_a$, denoted as $q(t_m)$ and $q(t_a)$ respectively, is a critical step that influences model performance. Specifically, $q(t_m)$ regularizes the core information

Table 11: Performance comparison (F1-score) on the stress test of the top 10% largest molecules versus the original random split.

| Modality | Model | f1-score (Top 10% Large) | f1-score (Original Split) |
|---|---|---|---|
| IR | 1D-CNN | 0.866 | 0.895 |
| | Transformer | 0.852 | 0.881 |
| | Wu et al. | 0.864 | 0.886 |
| | Alberts et al. | 0.874 | 0.891 |
| | **MSpecTmol** | **0.900** | **0.920** |
| $^{13}$C-NMR | 1D-CNN | 0.623 | 0.674 |
| | Transformer | 0.845 | 0.913 |
| | Wu et al. | 0.873 | 0.914 |
| | Alberts et al. | 0.896 | 0.919 |
| | **MSpecTmol** | **0.904** | **0.923** |
| IR + $^{13}$C-NMR + $^{1}$H-NMR | 1D-CNN | 0.873 | 0.900 |
| | Transformer | 0.902 | 0.936 |
| | Wu et al. | 0.912 | 0.944 |
| | Alberts et al. | 0.916 | 0.947 |
| | **MSpecTmol** | **0.925** | **0.959** |

extracted from the primary spectrum, while $q(t_a)$ regularizes the supplementary information from the auxiliary spectra. To ensure the scientific rigor and optimality of our model's configuration, we systematically investigated the impact of different prior distributions on performance.

We designed a series of rigorous comparative experiments to evaluate three distinct prior distributions on the functional group classification task, applying them to both $q(t_m)$ and $q(t_a)$:

1. **Gaussian Distribution**: The standard $\mathcal{N}(0, I)$ distribution.
2. **Laplace Distribution**: The standard $Laplace(0, 1)$ distribution, which is known to effectively promote sparsity in the latent space.
3. **Gamma Distribution**: The standard $\Gamma(k = 1, \theta = 1)$ distribution, which constrains the latent variables to be non-negative.

Throughout these experiments, all other model hyperparameters (such as learning rate, batch size, and the trade-off coefficients $\alpha$ and $\beta$) were held strictly constant to ensure a fair comparison.

The model trained with the **Gaussian prior** achieved the highest F1-score of 0.959, compared to 0.951 for the Laplace prior and 0.946 for the Gamma prior. This superior performance suggests that assuming the compressed latent features of both primary and auxiliary spectra follow a Gaussian distribution provides an efficient and flexible representation space, allowing the model to optimally capture the complex relationships within the spectral data. Therefore, we selected the Gaussian distribution for our final model configuration, as its effectiveness is validated by these results.

## M    MODEL PERFORMANCE STABILITY

To provide a more detailed view of model stability, Figure 6 visualizes the root-mean-square deviation (RMSD) distributions for conformations generated by different models. The MSpecTmol model exhibits a distribution with a notably lower median RMSD and smaller variance compared to all other baselines. This indicates that MSpecTmol not only generates conformations that are, on average, more accurate but also does so with higher consistency. In contrast, models relying on a single spectral modality (UV-Only, IR-Only, Raman-Only) and the attention-based ablation model all show higher median RMSDs and wider distributions. This demonstrates a greater variance in the quality of the generated conformations and underscores the effectiveness of our multi-modal fusion strategy in achieving stable, high-fidelity results.

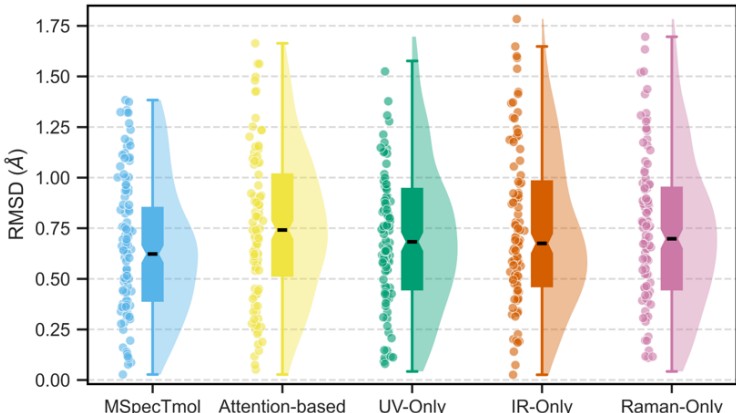

Figure 6: RMSD (Å) distributions for MSpecTmol compared to single-modality models and an attention-based ablation baseline. The black line indicates the median, the box represents the interquartile range, and the violin plot shows the probability density of the data.

## N    COMPARISON WITH ALTERNATIVE MULTI-MODAL FUSION STRATEGIES

To validate the effectiveness of PA-IB as a multi-modal information fusion strategy, we conducted a comparative study against three standard fusion paradigms:

- **Early Fusion:** The features from all modalities are directly concatenated at the input level before being fed into the model, allowing the model to learn a joint representation from the raw data.
- **Mid-level Fusion:** Each modality is first processed through independent CNN encoders to extract latent features. These features are then concatenated and passed to a shared MLP for classification.
- **Late Fusion:** Each modality independently predicts the presence of functional groups through separate MLPs, and the final prediction is obtained by averaging the probability outputs from all modalities.

The results, presented in Table 12, show that MSpecTmol consistently outperforms all three baselines (in terms of F1-score). Early and Mid-level fusion strategies generally perform better than Late fusion, likely because they enable some degree of joint representation learning. However, they still fall short of MSpecTmol, as they fail to explicitly filter out redundant cross-modal information. Late fusion performs the worst, as it ignores inter-modal interactions entirely by processing each modality in isolation, leading to substantial information loss.

Table 12: Performance comparison (F1-score) of MSpecTmol against Early, Mid-level, and Late fusion strategies across different modality combinations.

| Modality Configuration | MSpecTmol | Early Fusion | Mid-level Fusion | Late Fusion |
|---|---|---|---|---|
| IR + $^{13}$C-NMR + $^{1}$H-NMR | **0.959** | 0.900 | 0.904 | 0.874 |
| IR + MS/MS (Pos) + MS/MS (Neg) | **0.944** | 0.887 | 0.895 | 0.854 |

## O    INTERPRETABILITY ANALYSIS

To disentangle overlapping importance regions caused by functional group co-occurrence, we adopt a one-vs-all training strategy by training a dedicated model for each functional group. Each model receives the concatenation of all three spectral modalities as input. This design allows us to isolate the contribution of individual spectra to specific functional group predictions and analyze their region-wise importance, as shown in Figure 7.

Different spectral modalities emphasize distinct molecular features. Different types of spectroscopy capture different aspects of molecular structures. Infrared (IR) spectroscopy is particularly important in identifying functional groups such as carbonyl (C=O), hydroxyl (-OH), and amine ($-NH_2$). This is likely because IR spectroscopy primarily reflects the vibrational characteristics of polar functional groups, which exhibit strong absorption in the IR spectrum. In contrast, nuclear magnetic resonance (NMR) spectroscopy is more sensitive to structural motifs such as alkyl ($-CH_3$, $-CH_2-$), aromatic rings, and heterocycles. This is because NMR provides detailed insights into the electronic environment surrounding specific atomic nuclei, allowing for precise differentiation of these structural features. The complementary nature of these spectral modalities underscores the necessity of multi-modal approaches for comprehensive molecular characterization.

## P    CONFUSION MATRIX ANALYSIS

To investigate functional group misclassification, we construct a confusion matrix based on co-occurring prediction errors, counting the instances where two groups are simultaneously mispredicted for each test sample. As shown in Figure 8, this reveals that certain groups—notably Ether, Haloalkane, and Sulfide—are frequently confused. To diagnose this, we visualized the model's attention across the fused multi-modal spectra (Figure 7). The analysis demonstrates that these confusable groups exhibit significant overlapping attention, indicating that the model relies on shared features present across different spectral modalities for their identification. This finding highlights the inherent difficulty in distinguishing these groups, even when multiple sources of spectral information are available.

The observed spectral feature overlap is rooted in the intrinsic chemical properties of these functional groups. Similarities in their responses across various spectroscopic methods, such as shared absorption bands or related electronegativity profiles, create highly correlated features that are challenging to disentangle. This inherent ambiguity confirms that no single spectral modality contains sufficient information for perfect discrimination. It therefore becomes critical to employ a framework that can synergistically fuse complementary information from multiple spectra. Our approach is designed to address this very challenge, resolving ambiguities by integrating diverse spectral evidence to achieve more accurate classification.

## Q    IMPACT OF DATA AUGMENTATION

Real-world experimental spectra are often subject to variations from instrumental noise and calibration drift. This challenge is compounded by the scarcity of large-scale, multi-modal spectral datasets. Given that deep learning models, particularly those with convolutional neural network (CNN) architectures, are inherently data-hungry, data augmentation becomes a crucial technique. By synthetically expanding the training dataset to represent a wider range of experimental conditions, we can significantly enhance the model's generalization, robustness, and overall predictive performance. All experiments were conducted on molecular data obtained from the SDBS database.

We implemented and tested several augmentation strategies:

- **Horizontal Shift:** A random horizontal shift of up to 10 pixels is applied to the spectrum's data points.
- **Vertical Noise:** Uniform random noise (up to a level of 0.05) is added to the intensity values, with the noise magnitude being inversely scaled by the signal intensity.
- **Gaussian Smoothing:** A 1D Gaussian filter with a sigma value randomly chosen between 0.75 and 1.25 is applied to the spectrum.
- **Combined Strategies:** A horizontal shift or vertical noise is first applied, followed by the application of Gaussian smoothing.

As shown in Figure 10(a), all data augmentation strategies successfully improved the F1-score compared to the model trained on the original data (0.9134). Among these, the Horizontal Shift strategy was the most effective, achieving the highest F1-score of 0.9344. This suggests that teaching the model to be robust against positional variations in spectral peaks is highly beneficial. The addition of vertical noise also provided a substantial performance boost. Interestingly, while Gaussian smooth-

ing alone offered a modest improvement, combining it with other methods (e.g., vertical noise + smoothing) did not yield further gains and resulted in lower performance than the individual, more effective strategies. This indicates that while introducing variability is beneficial, excessive transformation can risk distorting the essential chemical information within the spectra, thereby creating a trade-off between robustness and signal fidelity.

## R  IMPACT OF MISSING MODALITIES AND NOISE INJECTION

**Impact of Missing Modalities**   To evaluate MSpecTmol's robustness against incomplete data, which is a common real-world challenge, we masked individual spectral modalities in the test set and evaluated the pre-trained model's performance on SDBS dataset. As presented in Table 13, we explicitly compared MSpecTmol against baseline models under these conditions.   As shown in Figure 10(b) and Table 13, the model's performance degrades gracefully rather than failing. The F1-score drops from 0.9134 with complete data to scores between 0.8974 and 0.8623 when a modality is absent. This resilience is a direct benefit of our PA-IB architecture. Notably, MSpecTmol with missing modalities still outperforms several baselines operating with complete data. The model's ability to extract information from auxiliary spectra to supplement the primary modality allows it to maintain robust performance even when data is partially available, making it highly suitable for practical applications.

Table 13: Performance comparison (F1-score) under missing modality conditions.

| Input Configuration | MSpecTmol | 1D-CNN | Trans. | Wu et al. | Alberts et al. |
|---|---|---|---|---|---|
| Full (MS+$^{13}$C+$^1$H) | **0.913** | 0.847 | 0.858 | 0.872 | 0.881 |
| w/o MS | **0.862** | 0.811 | 0.826 | 0.831 | 0.847 |
| w/o $^1$H-NMR | **0.897** | 0.823 | 0.834 | 0.848 | 0.866 |
| w/o $^{13}$C-NMR | **0.878** | 0.819 | 0.818 | 0.832 | 0.851 |

**Impact of Noise**   Noise is an unavoidable component of experimental spectra, making a model's performance under such conditions a key indicator of its practical utility. To assess this, we introduced varying levels of Gaussian noise to all spectra in the test set. The standard deviation of the noise was scaled proportionally to the maximum intensity of each spectrum, ensuring a consistent signal-to-noise ratio for the evaluation. As illustrated in Figure 10(c) and Table 14, the model's F1-score exhibits a steady and predictable decline as the noise level increases, decreasing from 0.9134 on clean data to 0.7469 at the highest noise level of 0.1. Importantly, the performance does not suffer a catastrophic collapse but rather degrades gracefully. Comparatively, as detailed in Table 14, baseline models suffer more severe degradation. At the highest noise level ($\sigma = 0.10$), MSpecTmol (0.747) significantly outperforms 1D-CNN (0.635) and Transformer (0.704).   This demonstrates that MSpecTmol can effectively discern core spectral features from background noise, further confirming its robustness for real-world applications where data quality is variable.

Table 14: Performance comparison (F1-score) under varying levels of Gaussian noise.

| Noise ($\sigma$) | MSpecTmol | 1D-CNN | Trans. | Wu et al. | Alberts et al. |
|---|---|---|---|---|---|
| 0.00 (Clean) | **0.913** | 0.847 | 0.858 | 0.872 | 0.881 |
| 0.02 | **0.882** | 0.815 | 0.825 | 0.838 | 0.846 |
| 0.05 | **0.825** | 0.745 | 0.761 | 0.767 | 0.772 |
| 0.10 | **0.747** | 0.635 | 0.704 | 0.711 | 0.703 |

## S  ALGORITHMIC PROCEDURE OF CONFORMATION GENERATION

In this work, we propose a dual-encoder diffusion framework that generates molecular conformations by conditioning a geometric diffusion model on spectroscopic information.

Table 15: Performance comparison of different models.

| Model | Top-1 Acc. | Top-5 Acc. | Top-1 mces-score | Top-5 mces-score |
|---|---|---|---|---|
| Transformer | 50.02% | 63.27% | 0.8674 | 0.9083 |
| Alberts et al. | 66.59% | 75.33% | 0.8821 | 0.9457 |
| Ours | **70.43%** | **85.25%** | **0.9469** | **0.9847** |

## S.1 PROBLEM FORMULATION

Given a molecular graph $G = (\mathcal{V}, \mathcal{E})$ with atom types $\mathbf{z} \in \mathbb{Z}^{|\mathcal{V}|}$ and spectroscopic measurements $\mathbf{s} = [\mathbf{s}_{uv}, \mathbf{s}_{ir}, \mathbf{s}_{raman}] \in \mathbb{R}^{d_s}$, we aim to generate 3D molecular conformations $\mathbf{x} \in \mathbb{R}^{3|\mathcal{V}|}$ that are consistent with both the molecular connectivity and observed spectra.

## S.2 SPECTROSCOPY-TO-SMILES PREDICTION

While it is technically feasible to generate complete 3D conformations directly from spectra, direct generation involves a huge potential space. An effective approach is to first obtain molecular samples based on spectra and then perform conformation generation. Since spectroscopy-to-SMILES prediction is already a well-established research direction, we conducted an additional feasibility study by replacing our predictor layer (MLP) with a Transformer-based decoder to predict SMILES strings directly from spectra. This modification does not alter our underlying PA-IB framework. To evaluate performance, we trained the model on the QM9S dataset—consistent with the conformation generation task—and calculated the Top-1 and Top-5 accuracy of the generated results, as well as the MCES score (normalized by the number of heavy atoms). The results are as Table 15:

These results show that our model achieves high accuracy on the dataset. Even in cases where predictions are not perfectly accurate, the MCES scores close to 1.0 demonstrate that the model is capable of reconstructing molecular graph structures with high similarity.

## S.3 SPECTRUM-CONDITIONED DIFFUSION PROCESS

We formulate the generation process as a conditional diffusion model operating in the coordinate space. The forward diffusion process adds Gaussian noise to the true conformation $\mathbf{x}_0$:

$$q(\mathbf{x}_t|\mathbf{x}_0) = \mathcal{N}(\mathbf{x}_t; \sqrt{\alpha_t}\mathbf{x}_0, (1 - \alpha_t)\mathbf{I}) \tag{29}$$

where $\alpha_t = \prod_{i=1}^{t}(1 - \beta_i)$ and $\{\beta_i\}$ follows a predefined noise schedule.

The reverse process is parameterized by a neural network $\boldsymbol{\epsilon}_\theta$ that predicts the noise conditioned on the spectrum:

$$\mathbf{x}_{t-1} = \frac{1}{\sqrt{1 - \beta_t}}\left(\mathbf{x}_t - \frac{\beta_t}{\sqrt{1 - \alpha_t}}\boldsymbol{\epsilon}_\theta(\mathbf{x}_t, t, \mathbf{s}, G)\right) + \sigma_t\boldsymbol{\eta} \tag{30}$$

where $\boldsymbol{\eta} \sim \mathcal{N}(0, \mathbf{I})$ and $\sigma_t$ is the posterior variance.

## S.4 DUAL-ENCODER ARCHITECTURE

Our model consists of three key components:

**Spectrum Encoder:** We design a MSpecTmol encoder to process multi-modal spectroscopic data. The spectrum data is encoded using the PA-IB-based method described previously. The features are fused through a gated mechanism:

$$\mathbf{h}_s = \text{MLP}(\mathbf{h}_{spec} \oplus \mathbf{t}_{emb}) \tag{31}$$

where $\mathbf{h}_{spec}$ is the spectrum embedding, $\mathbf{t}_{emb}$ is the timestep embedding, and $\oplus$ denotes concatenation.

**Dual Geometric Encoders:** We employ two complementary graph encoders: (1) a SchNet-based global encoder that captures long-range interactions through radius graphs, and (2) a GIN-based local encoder focusing on chemical bond structures. Both encoders incorporate the spectrum condition $\mathbf{h}_s$ into node representations.

**Distance-based Denoising:** In this step, we predict noise in the distance space and transform back to coordinates. The training objective combines global and local distance predictions:

$$\mathcal{L} = \mathbb{E}_{t,\boldsymbol{\epsilon}} \left[ \lambda_g \|\mathbf{d}_g - \hat{\mathbf{d}}_g\|_2^2 + \lambda_l \|\mathbf{d}_l - \hat{\mathbf{d}}_l\|_2^2 \right] \tag{32}$$

where $\mathbf{d}_g, \mathbf{d}_l$ are target distances for global and local edges respectively, and $\lambda_g, \lambda_l$ are weighting factors.

# T HYPERPARAMETER EXPERIMENTS AND ABLATION STUDY

To assess the impact of hyperparameter choices on model performance, we conduct a series of experiments by varying the information bottleneck trade-off coefficients $\alpha$ and $\beta$ from $1 \times 10^{-7}$ to $1 \times 10^{-3}$. The best results are achieved when both $\alpha$ and $\beta$ are set to $1 \times 10^{-6}$. Besides, we present the implementation details of the ablation settings and illustrate how the loss function changes when specific components are removed. The corresponding results are shown in Figure 11.

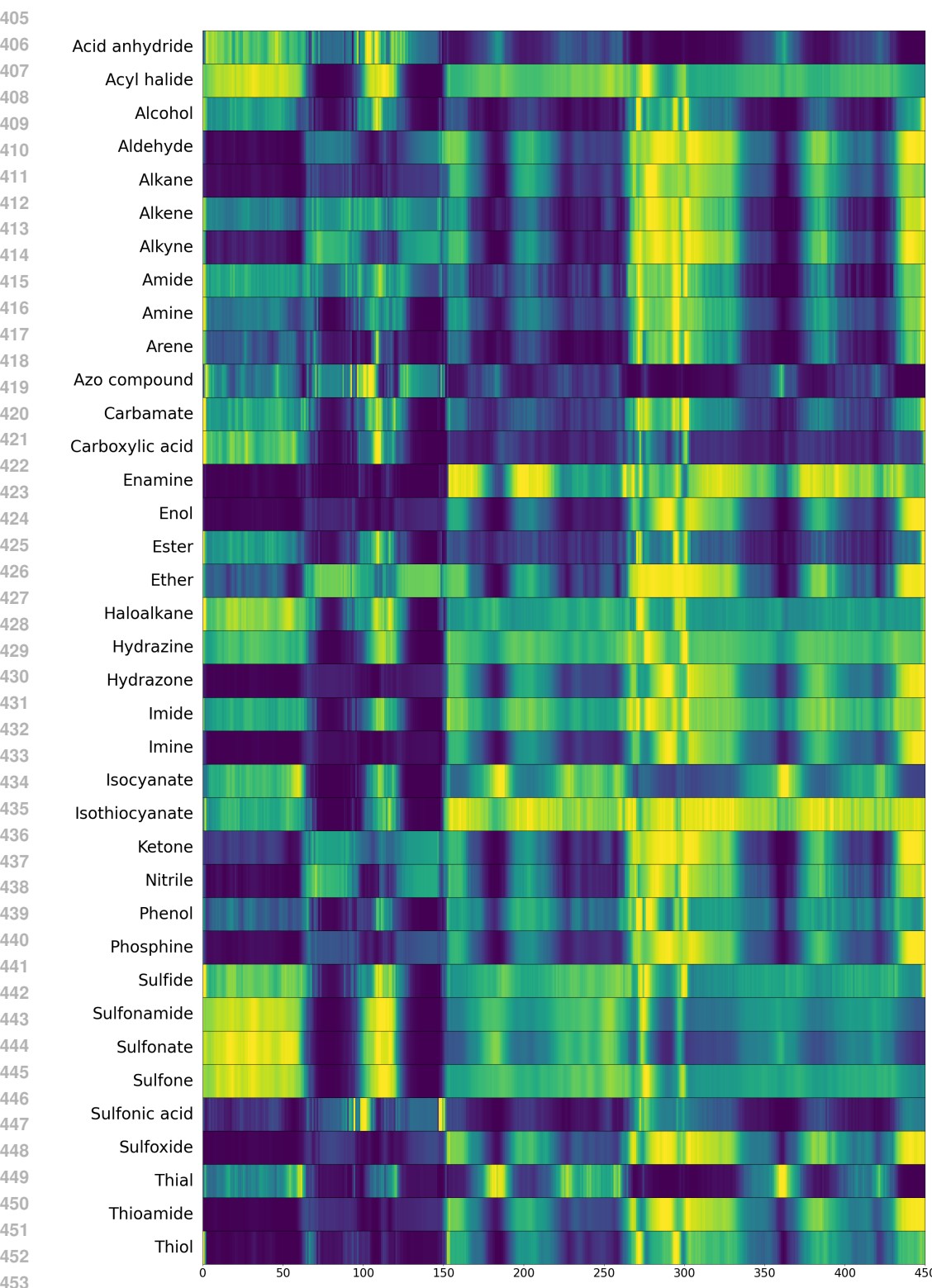

Figure 7: Illustration of the importance of spectral regions. The input spectrum is partitioned as follows: [0, 150] corresponds to IR spectra, [151, 300] to $^1$H-NMR spectra, and [301, 450] to $^{13}$C-NMR spectra. Warmer colors indicate crucial (high-importance) information, while cooler colors represent redundant (low-importance) information.

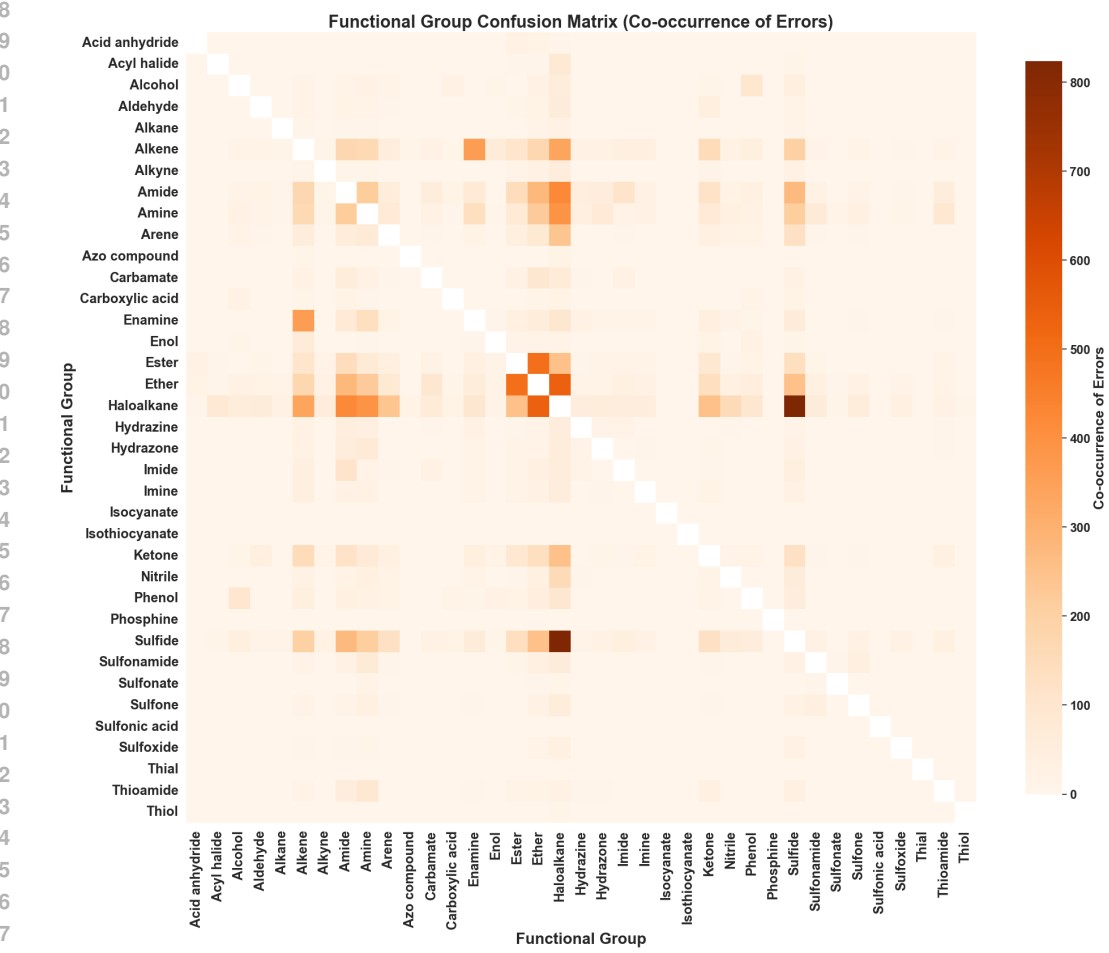

Figure 8: The confusion matrix between functional groups: the darker the color in the blocks, the higher the number of samples where the two functional groups were predicted incorrectly simultaneously.

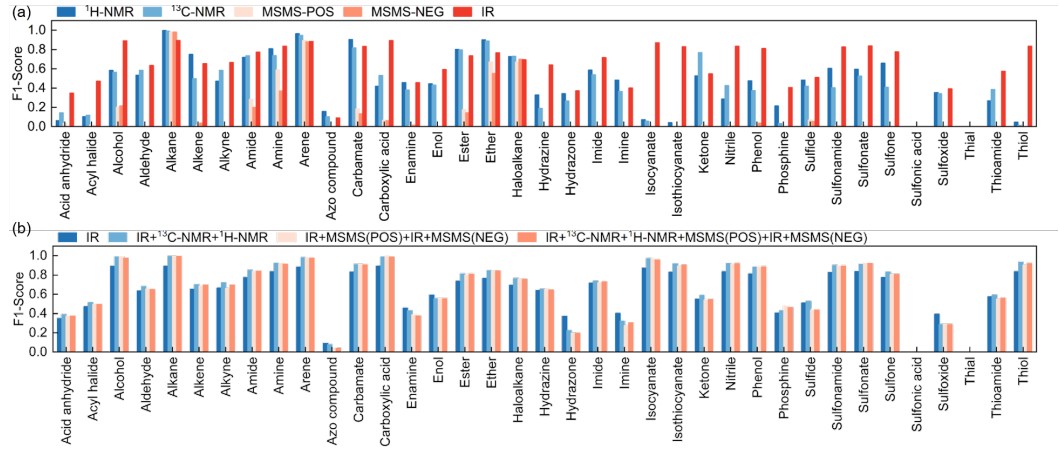

Figure 9: Performance of the model under unimodal and multimodal settings. (a) Results using a single spectrum as input (unimodal); (b) Results under multimodal fusion of spectra.

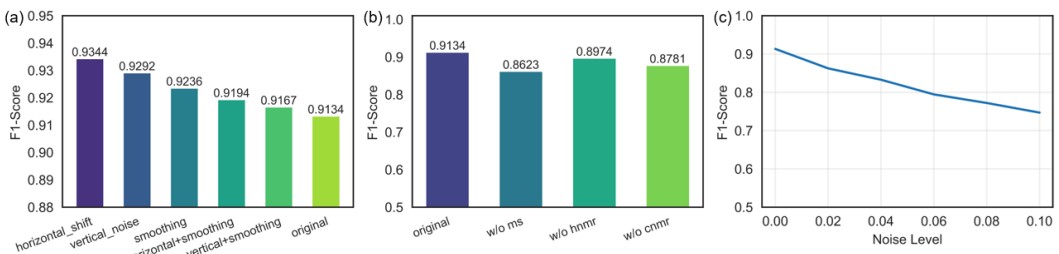

Figure 10: (a) Comparison of F1-scores for different data augmentation strategies applied to the training data. (b) Model performance with missing spectral modalities (MS, $^1$H-NMR, and $^{13}$C-NMR) in the test set. (c) The impact of increasing levels of Gaussian noise on the final F1-score.

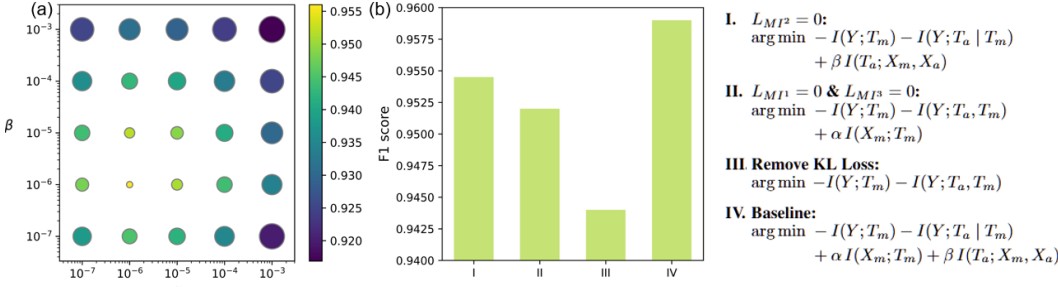

Figure 11: (a) Hyperparameter Experiments on functional group classification task. The circle size is proportional to the magnitude of the error. (b) Ablation study: by selectively removing different KL divergence terms, we adjust the optimization objectives of the model. The left panel shows the F1 scores of the prediction results, while the right panel illustrates the minimized objectives of the ablated models.

