# OpenReview forum: "MSpecTmol: A Multi-Modal Spectroscopic Learning  Framework for Molecular Structure Identification"
_ICLR.cc/2026/Conference — ICLR 2026 Conference Withdrawn Submission_

### Official Review · Reviewer_9qDe · 2025-10-23

**Soundness:** 1
**Presentation:** 2
**Contribution:** 2
**Rating:** 2
**Confidence:** 4

**Summary:**

The spectroscopic techniques play an important role in structural determination for an unknown molecule. Current challenges. 1) Fully leveraging the available spectroscopic data to extract the physical structure and to make structural determination more accurate; Previous methods only use low-dimensional features and restricted information. Novelties. 1) A primary-auxiliary synergistic modeling approach to capture both core information and supportive information.Results. 1) F1 score of 0.959 on both simulated and experimental spectra; 2) Avg. RMSD 0.682 A on the spectrum-conditioned conformation generation task. 3) Align well with the chemical intuition.

**Strengths:**

1. This paper has calculated the theoretical upper bounds of each objective as the loss function, which makes the optimization easier.
2. MSpecTmol has designed a specific optimization function for spectrum detection with the two modalities.
3. The experiment results show consistent advantages on selected tasks.

**Weaknesses:**

1. The motivation of MSpecTmol is not crucial. The key claim is that other models only use individual spectroscopic, which didn't consider models like DffSpectra, SpectraLLM, MMST, and older models such as spec2struct. As a result, the motivation of "using multi-spectra" is not solid here. The main challenge you solve may be "how to integrate multiple spectrums better. In this case, you should compare similar multimodal models.
2. For the classification tasks, the F1 score is not enough, especially for imbalanced datasets. The paper should show more metrics under this reason, and analyze the dataset distribution
3. The design ofthe primary-auxiliary encoding module may be sensitive to the primary modality. The paper didn't analyze the effectiveness of the symmetric fusion for these two modalities. The selection of modalities is also important , such as why we select the IR as the main modality. In conclusion, the paper's ablation study is still weak, leak the design rational of each module in the framework.4.Some writing tips: the equation (9) is a repeatition of (1), you can just reference equation (1); the font in the figure should be consistent with that in the main text, it is better to use selectable text format in the figure (e.g. using pdf format for figure), otherwise some tiny text can not be seen in the figure.

**Questions:**

1. For Table 4 (line 934), when you compare the resource usage, did you use the same model size for different models? If the model sizes vary a lot, can you specify them?
2. Where did you cite Table 5? What are the 5 modalities in these experiments? The resource usage looks totally acceptable for 5 modalities with a better performance (only a normal commercial GPU can handle this, no need to trade off). Why not use all the modalities?
3. Why do you use a CNN to model the original data? And why MLP for importance score, why self-attention for fragment extraction? Did you do an ablation for each module in the framework?
4. Why did you select the IR spectra as the primary modality and the other two as auxiliary? Did you do an ablation study to prove your selection?

---

> ### Author Response · Authors · 2025-11-22
> **Response part I**
>
> We sincerely thank the reviewer for acknowledging the theoretical foundations and consistent experimental advantages of our work. We have carefully addressed all weaknesses and questions below.
>
> >  **W1. Motivation and Comparison with Multi-Modal Models**
>
> We appreciate the reviewer’s feedback and would like to further clarify the novelty and motivation of our approach.
>
> First, we agree that simply stating that “multi-spectra contain richer information” is not a sufficient motivation on its own. However, our goal is not to suggest that prior works fully ignored multi-spectral inputs. Rather, our central motivation is that existing models (e.g., DiffSpectra, SpectraLLM) treat each modality either independently or via shallow feature concatenation, and therefore fail to address the core challenge: the difficulty is not in using multiple spectra, but in integrating heterogeneous spectroscopic signals in a physically coherent and complementary manner.
>
> Beyond proposing the PA-IB framework, our contributions extend further. PA-IB performs principled asymmetric information compression that **enables effective extraction of complementary spectral cues**. This design yields substantial improvements in molecular identification accuracy, and importantly **allows high-quality 3D structure reconstruction from spectra**, which has not been explored in prior methods. In addition, our model **provides clear and chemically meaningful interpretability** by highlighting spectral regions that contribute most to structural inference.
>
> To better address the reviewer’s concern, we have additionally:
>
> * **Conducted new ablation studies to further validate the effectiveness of the PA-IB framework**
>
> To validate the effectiveness of our asymmetric approach, we conducted an ablation study comparing PA-IB against a symmetric baseline that applies uniform IB constraints across all modalities. Specifically, we compared the following two objectives:
>
> 1） *PA-IB (Ours):* $-I(Y; T_m) - I(Y; T_a | T_m) + \alpha I(X_m; T_m) + \beta I(T_a; X_m, X_a)$
>
> 2）*Uniform IB:* $-I(Y; T_m, T_a) + \alpha I(X_m; T_m) + \beta I(X_a; T_a)$
>
> The Uniform IB baseline compresses each modality independently, without considering the complementary relationship between modalities or addressing cross-modal redundancy.
>
> | Objective        | IR        | $^{13}$C-NMR | $^1$H-NMR | IR + $^{13}$C-NMR + $^1$H-NMR |
> | ---------------- | --------- | ------------ | --------- | ----------------------------- |
> | **PA-IB (Ours)** | **0.923** | **0.920**    | **0.927** | **0.959**                     |
> | Uniform IB       | 0.906     | 0.904        | 0.905     | 0.934                         |
>
> As shown in the table above, PA-IB consistently outperforms the Uniform IB baseline across all configurations, with a 2.5% absolute improvement in the multi-modal case (0.959 vs. 0.934). The Uniform IB baseline suffers from redundancy across modalities, which illustrates the limitations of symmetric fusion. This performance gap is particularly pronounced in multi-modal settings, emphasizing that the asymmetric hierarchical design of PA-IB offers unique advantages that standard IB regularization cannot achieve.
>
> * **Compared multiple fusion strategies**
>
> To validate its effectiveness as a multi-modal information fusion strategy, we conducted an ablation study comparing PA-IB with models that use three different information fusion strategies:
>
> * **Early Fusion**: The features from all three modalities are directly concatenated before being input into the model, allowing the model to learn a joint representation from the raw data.
> * **Mid-level Fusion**: Each modality is processed through CNNs to extract features, which are then concatenated and used as input features for an MLP.
> * **Late Fusion**: Each modality independently predicts the presence of functional groups through an MLP, and the final prediction is obtained by averaging the predictions from all modalities.
>
> | Modality                      | MSpecTmol | Early Fusion | Mid-level Fusion | Late Fusion |
> | ----------------------------- | --------- | ------------ | ---------------- | ----------- |
> | IR + $^{13}$C-NMR + $^1$H-NMR | 0.959     | 0.900        | 0.904            | 0.874       |
> | IR + MS/MS_pos + MS/MS_neg    | 0.944     | 0.887        | 0.895            | 0.854       |
>
> From the results, MSpecTmol superior three fusion baselines (F1-score index). Early fusion and mid-level fusion perform better than late fusion, likely because they provide some level of joint representation learning but still fail to remove redundant cross-modal information. Late fusion performs the worst since it ignores inter-modal interactions entirely, processing each modality independently and merely averaging predictions, which leads to substantial information loss.

---

> ### Author Response · Authors · 2025-11-22
> **Response part II**
>
> * **Incorporated the additional baselines suggested by the reviewer.**
>
> The baselines you mentioned are indeed advanced models; however, each of them has certain limitations as follows:
>
> 1) DiffSpectra is indeed highly relevant to our problem setting, as it also aims to predict molecular conformations from multiple spectral modalities. However, there are several reasons why it cannot be directly included as a baseline in our experiments.  **First, DiffSpectra is available only as an arXiv preprint**, and its official implementation has not been released. Despite our efforts, **we were unable to reliably reproduce its results, and therefore we chose to take a conservative position and avoid reporting potentially inaccurate numbers**.  Second, the **modeling objectives and generation pipelines of the two approaches differ fundamentally**. DiffSpectra performs direct end-to-end conformation generation, yielding an RMSD around 1.6 Å. In contrast, our method adopts a two-stage paradigm that first reconstructs a molecular graph and then achieve conformations generation, achieving significantly lower RMSD (≈ 0.68 Å). Because the underlying workflows and evaluation settings are not aligned, a direct numerical comparison would be misleading.  For these reasons, we believe that including DiffSpectra as a baseline would not provide a fair or meaningful comparison. Nonetheless, we acknowledge its relevance and will mention it in the related work section of the revised manuscript.
>
>
> 2) SpectraLLM performs molecular structure analysis through large language models. However, **this work is currently only available on arXiv, and no reproducible code has been released**. Thus, its results cannot be used as reliable references for comparison.
>
> 3) MMST is also an **arXiv preprint, and it heavily relies on 2D-NMR modalities such as HSQC and COSY**. When these modalities are removed, its performance degrades significantly. More importantly, the work is designed around these high-dimensional NMR experiments, making its setting incompatible with ours.
>
> 4) spec2struct is a model that generates molecular structures through a retrieval-based mechanism. Its performance largely depends on the database it uses rather than generalizable molecular prediction. More importantly, it uses the Alberts et al. dataset—the same dataset as our training and testing—so comparing with it would not be fair, as **retrieval-based methods benefit greatly from database overlap rather than true predictive ability**.
>
> However, the above methods are important research advances in the field of AI for spectroscopy, and we will discuss them in the **Related Work section**. Meanwhile, for the conformation generation task, **we add a set of reliable and widely recognized baselines as follows**:
>
> To validate the effectiveness of this approach, we conducted a clear and fair comparison with existing graph-only methods. We directly compared our spectrum-conditioned model against baselines that generate conformations solely from graphs, including rule-based methods (RDKit ETKDGv3 and OpenBabel) and deep learning-based methods (GeoDiff). To ensure a fair comparison, we selected the top-5 conformations for each model and calculated the average RMSD for the target conformation.
>
> | Method                      | Condition            | Avg. RMSD (Å) |
> | :-------------------------- | :------------------- | :------------ |
> | RDKit                       | Graph or SMILES      | 1.350         |
> | OpenBabel                   | Graph or SMILES      | 1.279         |
> | GRAPHDG                     | Graph or SMILES      | 1.247         |
> | CGCF                        | Graph or SMILES      | 1.246         |
> | GEOMOL                      | Graph or SMILES      | 1.175         |
> | CONFGF                      | Graph or SMILES      | 1.143         |
> | JODO                        | Graph or SMILES      | 1.124         |
> | GeoDiff                     | Graph or SMILES      | 1.074         |
> | Ours (Single Modality)      | Spectrum + Graph     | 0.697         |
> | **Ours (Three Modalities)** | **Spectrum + Graph** | **0.682**     |
>
> We hope these clarifications and the newly added experiments sufficiently address the reviewer’s concerns.

---

> > ### Author Response · Authors · 2025-11-22
> > **Response part III**
> >
> > >  **W2. Comprehensive Evaluation Metrics and Dataset Analysis**
> >
> > We acknowledge the reviewer's concern regarding the limitations of the F1 score in imbalanced settings. In Table 1, we primarily employed the Micro-F1 score because it effectively captures the model's overall precision and recall capabilities on large-scale datasets[1]. However, to further evaluate the impact of the class imbalance phenomenon, we truly introduced several additional metrics and analyses as follows:
> >
> > | **Model**    | **MSpecTmol** | **Alberts et al.** | **Wu et al.** | **1D-CNN** | **Transformer** |
> > | :----------- | :-----------: | :-------: | :-------: | :--------: | :-------------: |
> > | **Macro F1** |     0.809    |       0.694       |     0.664    |   0.608   |      0.587     |
> > | **ACC**      |     0.726    |       0.663       |     0.634    |   0.598   |      0.397     |
> > | **Micro F1** |     0.959    |       0.947       |     0.944    |   0.936   |      0.900     |
> >
> >
> > In terms of ***Dataset distribution:*** We have explicitly visualized the functional group number distribution in the test set in Figure 3d, and here we present the detailed statistical results.
> >
> > * **Functional Group Distribution**
> >
> > | Functional Group   |   Count | Functional Group   | Count   | Functional Group   | Count   | Functional Group   | Count   |
> > |---------|--------|-------------|--------|---------|--------|------|--------|
> > | Alkane |  765130 | Arene      | 615081  | Ether     | 449323  | Haloalkane         | 366179  |
> > | Amine    |  335867 | Alcohol   | 213059  | Amide   | 209647  | Ester  | 149437  |
> > | Sulfide   |  108004 | Alkene     | 91781   | Carboxylic acid    | 79421   | Carbamate   | 76183   |
> > | Ketone    |   68771 | Nitrile    | 52403   | Phenol     | 46250   | Sulfonamide    | 36486   |
> > | Aldehyde   |   20323 | Sulfone    | 19049   | Alkyne     | 15223   | Imide  | 14787   |
> > | Enamine   |   11168 | Sulfonate   | 9014    | Hydrazine    | 8812    | Thioamide          | 6033    |
> > | Imine   |    5950 | Hydrazone  | 5413    | Sulfoxide          | 4546    | Acyl halide        | 4073    |
> > | Thiol    |    2864 | Enol  | 2552    | Isocyanate         | 922     | Isothiocyanate     | 738     |
> > | Phosphine    |     651 | Acid anhydride     | 618     | Azo compound       | 542     | Thial              | 48      |
> > | Sulfonic acid      |      10 |
> >
> > * **Heavy Atom Distribution**
> >
> > | Range |   Count | Percentage |
> > |--------|--------|--------|
> > | 5-10    |   22404 | 2.82%    |
> > | 11-15   |  126376 | 15.91%  |
> > | 16-20   |  180571 | 22.73% |
> > | 21-25   |  179398 | 22.58% |
> > | 26-30   |  166491 | 20.96%   |
> > | 31-35   |  119163 | 15.00%  |
> >
> > * **Sample-wise Functional Group Count Distribution**
> >
> > | Range   |   Count | Percentage   |
> > |--------|--------|-------------|
> > | 0-2     |   34706 | 4.37%  |
> > | 3-4     |  306620 | 38.60%   |
> > | 5-6     |  369813 | 46.55%  |
> > | 7-8     |   78108 | 9.83%  |
> > | >=9     |    5156 | 0.65%   |
> > We hope the above results can well address your concerns.
> >
> > >  **W3.  Symmetric Fusion Analysis**
> >
> > Beyond proposing the PA-IB framework, our contributions extend further. PA-IB performs principled asymmetric information compression that enables effective extraction of complementary spectral cues. This design yields substantial improvements in molecular identification accuracy, and—importantly—allows high-quality 3D structure reconstruction from spectra, which has not been explored in prior methods. In addition, our model provides clear and chemically meaningful interpretability by highlighting spectral regions that contribute most to structural inference.
> >
> > To better address the reviewer’s concern, we have additionally:
> >
> > * **Conducted new ablation studies to further validate the effectiveness of the PA-IB framework**
> >
> > To validate the effectiveness of our asymmetric approach, we conducted an ablation study comparing PA-IB against a symmetric baseline that applies uniform IB constraints across all modalities. Specifically, we compared the following two objectives:
> >
> > - **PA-IB (Ours):** $-I(Y; T_m) - I(Y; T_a | T_m) + \alpha I(X_m; T_m) + \beta I(T_a; X_m, X_a)$
> > - **Uniform IB:** $-I(Y; T_m, T_a) + \alpha I(X_m; T_m) + \beta I(X_a; T_a)$
> >
> > The Uniform IB baseline compresses each modality independently, without considering the complementary relationship between modalities or addressing cross-modal redundancy.
> >
> > | Objective        | IR        | $^{13}$C-NMR | $^1$H-NMR | IR + $^{13}$C-NMR + $^1$H-NMR |
> > | ---------- | --------- | ------------ | --------- | -------------- |
> > | **PA-IB (Ours)** | **0.923** | **0.920**    | **0.927** | **0.959**  |
> > | Uniform IB       | 0.906     | 0.904   | 0.905     | 0.934  |
> >
> > As shown in the table, PA-IB consistently outperforms the Uniform IB baseline, with a 2.5% absolute improvement in the multi-modal case (0.959 vs. 0.934).This performance gap is particularly pronounced in multi-modal settings, emphasizing that the asymmetric hierarchical design of PA-IB offers unique advantages that standard IB regularization cannot achieve.

---

> > > ### Author Response · Authors · 2025-11-22
> > > **Response part IV**
> > >
> > > >  **W4.  Equation 9 the equation (9) is a repeatition of (1),  the font in the figure should be consistent with that in the main text**
> > >
> > > We clarify that Equation (1) formulates the theoretical objective of the PA-IB design, whereas Equation (9) specifies the practical optimization objective used for model training. Although they share the same mathematical form, they serve distinct narrative roles: the former establishes the design principle, while the latter guides the implementation. This mathematical identity is intentional, highlighting the strict alignment between our theoretical framework and the actual model realization.
> > >
> > > As for the font usage, while there are no explicit requirements for figure fonts in the ICLR style guide, we appreciate your suggestion for visual consistency. We will unify the fonts in the figures following your suggestions.
> > >
> > >
> > > >  **Q1. Model Size Comparison in Table 4**
> > >
> > > We agree that forcing a "same size" constraint is inappropriate given the fundamental architectural differences. Overall, the size differences among these models are not substantial, and their parameter counts are broadly comparable. For transparency, we provide the parameter counts corresponding to the results in Table 4:
> > >
> > > | Model | Parameters | Memory (GB) | Training Time (h) | F1-score |
> > > | :--- | :--- | :--- | :--- | :--- |
> > > | 1D-CNN | 47.8M | 5.7 | 2 | 0.900 |
> > > | Transformer | 30.4M | 1.7 | 35 | 0.936 |
> > > | MSpecTmol | 57.4M | 6.6 | 2.5 | 0.959 |
> > >
> > > MSpecTmol’s size is justified by its superior performance (F1=0.959) and exceptional efficiency.
> > >
> > > >  **Q2. Where did you cite Table 5? What are the 5 modalities in these experiments?**
> > >
> > > **Table 5 Context:** Table 5, located in **Appendix G**, is used to analyze the scalability and complexity trade-off as the number of input modalities increases.
> > >
> > > **The 5 Modalities:** The five modalities tested (based on the Alberts et al. dataset) are: **IR, $^{13}$C-NMR, $^1$H-NMR, MS/MS (positive mode), and MS/MS (negative mode)**.
> > >
> > >
> > > >  **Q2. Why not use all the modalities？**
> > >
> > > Increasing the input from 3 to 5 modalities resulted in diminishing returns (F1 improvement of only 0.4%, from 0.959 to 0.963), while resource consumption nearly doubled (Memory: 6.6GB $\to$ 11.3GB; Training Time: 2.5h $\to$ 5h). We selected the 3-modality configuration (IR, $^{13}$C-NMR, $^1$H-NMR) as the standard because it provides the **optimal balance** between predictive performance and computational feasibility, aligning with the high practical cost of acquiring multiple spectra in real-world scenarios.

---

> > > > ### Comment · Reviewer_9qDe · 2025-11-25
> > > > **Reply to Q2 Response**
> > > >
> > > > - For the Table 5 issue, I found Table 5 is in Appendix G. Actually, when I wanted to find the related explanation text of Table 5, I failed. So it is important that you explicitly reference each table and figure in writing, so that the reader can glance at a table and find the related explanation easily.
> > > > - For the training time calculation, I am concerned about the number: when the memory usage doubled, the training time doubled as well. Can you provide more evidence on what hyperparameter settings can cause this phenomenon? Such as can you compare the model size, batched data size, and training flops? As well, can you provide the information about the hardware settings, such as the GPU model, float precision, so that I can reproduce and check the training efficiency?
> > > >
> > > > Thank the authors for their hard work, but I think these concerns are important to address.

---

> > > ### Comment · Reviewer_9qDe · 2025-11-25
> > > **Replay to Repsonse Part 3**
> > >
> > > Thank the authors for these added metrics. I have searched online, the Macro-F1 and Micro-F1 are enough to show the performance on imbalanced data. And thank the author for the distribution analysis, which is a good habit for evaluating datasets of this kind.

---

> > ### Comment · Reviewer_9qDe · 2025-11-25
> > **Reply to Response Part 2**
> >
> > Thank the authors for the rebuttal. The author said the baselines' codes are not released. However, I just searched and found the official codes for the DiffSpectra and MolSpectLLM (which the author said it is only a preprint and has not released their code)
> >
> > https://github.com/AzureLeon1/DiffSpectra
> > https://github.com/Eurekashen/MolSpectLLM
> >
> > Can you explain why you said their code is not publicly available?
> >
> > Your baselines in this table are all only graph or SMILES; is there really not a multimodal spectrum model to compare?
> >
> > I would appreciate it if the authors could conduct a further investigation and a detailed analysis.

---

> ### Author Response · Authors · 2025-11-22
> **Response part V**
>
> >  **Q3. Reason for Architectural Components (CNN, MLP, Self-Attention)**
>
> Our architectural choices are guided by the properties of spectral data and the requirements of the PA-IB framework:
>
> *   **1D-CNN for Encoding:** Spectral data is an **inherently 1D sequence** containing local patterns (e.g., specific peaks and peak shapes). 1D-CNNs are used to leverage **translation-invariant feature extraction**, effectively capturing these low-level local motifs. These are common methods used by current models for processing spectral data.[1,2]
>
> *   **MLP for Importance Scoring:** The MLP serves as an **efficient non-linear function approximator**. Its role is to map the high-dimensional latent features to the scalar probabilities ($p_{mi}$, $p_{ai}$ in Eq. 6) needed to quantify the importance of each frequency band for the information bottleneck compression. These are common methods used by AI to determine importance.[3,4]
>
> *   **Self-Attention for Fragment Extraction:** After the PA-IB module filters out redundancy, the self-attention mechanism is employed to integrate **long-range dependencies**. This is essential for correlating distant, yet chemically linked, spectral features across the entire spectrum. [5,6]
>
> While these components represent standard methodologies in modern spectral representation learning, our core contribution lies in the proposal of the PA-IB framework. By performing principled asymmetric information compression, PA-IB effectively extracts complementary spectral cues. This design not only yields substantial improvements in molecular identification accuracy but also pioneers high-quality 3D structure reconstruction from spectra. Furthermore, our model offers clear, chemically meaningful interpretability by highlighting the spectral regions that contribute most to structural inference. **Therefore, what is important is the design of the model algorithm and the construction of the framework, rather than the mere use of modules.**
>
>
> [1] Jung, G., Jung, S. G., & Cole, J. M. (2023). Automatic materials characterization from infrared spectra using convolutional neural networks. *Chemical Science*, 14(13), 3600-3609.
>
> [2] Alberts, M., et al. (2024). Unraveling molecular structure: A multimodal spectroscopic dataset for chemistry. *Advances in Neural Information Processing Systems*, 37, 125780-125808.
>
> [3] Lee N, Hyun D, Na G S, et al. (2023) Conditional graph information bottleneck for molecular relational learning. *International Conference on Machine Learning*. PMLR, 2023: 18852-18871.
>
> [4] Bahdanau D, Cho K, Bengio Y(2015). Neural machine translation by jointly learning to align and translate. *International Conference on Learning Representations*.
>
> [5] Wu, W., et al. (2025). Transformer-Based Models for Predicting Molecular Structures from Infrared Spectra Using Patch-Based Self-Attention. *The Journal of Physical Chemistry A*, 129(8), 2077-2085.
>
> [6] Alberts M, Zipoli F, Laino T (2025). Setting new benchmarks in AI-driven infrared structure elucidation. *Digital Discovery*.
>
> >  **Q4. Primary Modality Selection**
>
> Thanks for your advice, in fact, Our choice of IR as the primary modality is data-driven and follows the principled strategy outlined in Appendix G.
>
> *  Single-modality Performance: Single-modality evaluation (Table 1 and Figure 9a) showed that IR spectroscopy delivered the highest and most stable prediction performance (F1=0.923). It is thus best suited to carry the core predictive information.
>
> *  Modality Selection Ablation: Table 1 explicitly compares multi-modal performance when selecting different primary inputs: IR as primary yielded the best result (F1 = 0.959) compared to $^{13}$C-NMR (0.957) or $^1$H-NMR (0.956) as primary.
>
> * Chemical Intuition: From a physicochemical perspective, IR spectroscopy is fundamentally the most direct modality for this task because it measures molecular vibrations[1] that act as distinct "fingerprints" for functional groups. High-polarity bonds found in key functional groups exhibit strong dipole moment changes, resulting in intense, diagnostic absorption peaks in the IR spectrum. In contrast, NMR ($^{1}$H and $^{13}$C) primarily elucidates the carbon skeleton and local atomic environments.
>
> For the reasons mentioned above, we have adopted infrared as the main modality, and we will enhance the relevant explanations in the Appendix.
>
> [1] Wilson E B, Decius J C, Cross P C. Molecular vibrations: the theory of infrared and Raman vibrational spectra[M]. Courier Corporation, 1980.
>
> ---
>
> We greatly appreciate your insightful and helpful comments, as they will undoubtedly help us improve the quality of our article. If our response has successfully addressed your concerns and clarified any ambiguities, we respectfully hope that you consider raising the score. Should you have any further questions or require additional clarification, we would be delighted to engage in further discussion.

---

> > ### Comment · Reviewer_9qDe · 2025-11-25
> > **Reply to Response Part 5**
> >
> > - Thank the authors for introducing 1dCNN to the self-attention. I know that the paper's novelty comes from the pipeline instead of the modules. But a better module definitely can improve the overall performance further. As there are a lot of MLP modules inside the pipeline which I think it's worth a try to use a transformer model, which is much stronger for embedding tasks. To prove your pipeline is good enough and not rely on any of these modules, at least a module ablation should be conducted, but this rebuttal does not have this.
> > - I am not sure whether it is my browser's problem; the equations in Q4 are not displayed correctly. If not, it could indicate that the authors did not check their rebuttal content before submitting.
> > - Also, ablation on primary/auxiliary modalities is still needed. Can you do the ablation, or cite someone else who did the ablation (not only cite the paper, please specify which table I can check) to prove your choice?

---

> > > ### Author Response · Authors · 2025-11-27
> > > **Response to Further question for Part V**
> > >
> > > > *MLP>>Transformer*
> > >
> > >
> > > Following your suggestions, we conducted a series of ablation studies to evaluate the architectural choices regarding the MLP and CNN components within our model.
> > >
> > > **Data Modeling**
> > >
> > > In our original framework, 1D-CNNs are employed to extract features from continuous spectral signals. To investigate the efficacy of a pure Transformer approach in this module, we discretized the continuous spectra into textual token sequences. We then replaced the 1D-CNNs with Embedding layers, directly feeding the token embeddings into the Transformer encoder to capture core spectral features.
> > >
> > > **Importance Score Acquisition**
> > >
> > > In our PA-IB module, an MLP is used to process spectral features and generate importance scores (gating probabilities). We retained this design as it is the standard methodology in Information Bottleneck literature ([1]- in Section 4.3.2,[2]-in Section 4.1,[3]-in Equation 6). MLPs are theoretically justified for mapping latent representations to distribution parameters due to their universal approximation capabilities. Therefore, we consider this module scientifically sound.
> > >
> > > **Functional Group Prediction (Output Head)**
> > >
> > > Our original model uses an MLP to map flattened features to the final functional group predictions. To replace this with a Transformer-style architecture, we introduced a learnable Query Token to perform Cross-Attention with the compressed spectral feature sequence, aggregating global information before passing it through a single linear layer for the final prediction.
> > >
> > > The results are presented below:
> > >
> > > | **Module Variant**                 | **F1-score** | **Macro-F1** |
> > > | ---------------------------------- | ------------ | ------------ |
> > > | **Original (CNN + MLP)**           | **0.959**    | **0.810**    |
> > > | Replace Data Modeling (Embedding)  | 0.947        | 0.713        |
> > > | Replace Output Head (Attn Pooling) | 0.937        | 0.625        |
> > >
> > >
> > >
> > > As shown in the table, the original architecture (CNN + MLP) outperforms the Transformer-variant replacements. This suggests that: (1) **the inductive bias of 1D-CNNs is highly effective for spectral data processing** compared to token-based embeddings, which may lose precise intensity information during discretization. (2) **MLP is more effective at capturing the complex relationships within the spectral latent space** compared to the Attention Pooling mechanism. These findings validate our original design choices as the most effective configuration for this specific task.
> > >
> > >
> > >
> > > [1] Lee N, Hyun D, Na G S, et al. Conditional graph information bottleneck for molecular relational learning[C]//International Conference on Machine Learning. PMLR, 2023: 18852-18871.
> > >
> > > [2] Yu J, Cao J, He R. Improving subgraph recognition with variational graph information bottleneck[C]//Proceedings of the IEEE/CVF conference on computer vision and pattern recognition. 2022: 19396-19405.
> > >
> > > [3] Zhang S, Fang J, Li X, et al. Iterative Substructure Extraction for Molecular Relational Learning with Interactive Graph Information Bottleneck[C]//The Thirteenth International Conference on Learning Representations. 2025.
> > >
> > > >  *Q4 are not displayed correctly.*
> > >
> > > Thank you for pointing out the display issue. We suspect you are referring to the **${}^{13}\text{C-NMR}$** and **${}^{1}\text{H-NMR}$** notation, which is the only equation-like element in Q4, and we have verified it displays correctly on our end. If the problem persists on your side, please let us know so we can revise the formatting for better compatibility.

---

> > > > ### Author Response · Authors · 2025-11-27
> > > > **Response to Further question for Part V (2)**
> > > >
> > > > > *Ablation on primary/auxiliary modalities is still needed.*
> > > >
> > > > Following your suggestion, we conducted ablation studies to evaluate different configurations of primary and auxiliary modalities. The results are presented below:
> > > >
> > > > | **Primary Modality** | **Auxiliary Modality**              | **F1-score** |
> > > > | -------------------- | ----------------------------------- | ------------ |
> > > > | IR                   | -                                   | 0.923        |
> > > > | IR                   | ${}^{13}$C-NMR                      | 0.930        |
> > > > | IR                   | ${}^{1}$H-NMR                       | 0.941        |
> > > > | **IR**               | **${}^{13}$C-NMR + ${}^{1}$H-NMR**  | **0.959**    |
> > > > | ${}^{13}$C-NMR       | -                                   | 0.920        |
> > > > | ${}^{13}$C-NMR       | IR                                  | 0.928        |
> > > > | ${}^{13}$C-NMR       | ${}^{1}$H-NMR                       | 0.935        |
> > > > | ${}^{13}$C-NMR       | IR + ${}^{1}$H-NMR                  | 0.957        |
> > > > | ${}^{1}$H-NMR        | -                                   | 0.927        |
> > > > | ${}^{1}$H-NMR        | IR                                  | 0.933        |
> > > > | ${}^{1}$H-NMR        | ${}^{13}$C-NMR                      | 0.944        |
> > > > | ${}^{1}$H-NMR        | IR + ${}^{13}$C-NMR                 | 0.956        |
> > > > | -                    | IR + ${}^{13}$C-NMR + ${}^{1}$H-NMR | 0.934        |
> > > >
> > > > The results demonstrate that the configuration using IR as the primary modality with ${}^{13}$C-NMR and ${}^{1}$H-NMR as auxiliary modalities achieves the highest performance. This suggests that the IR spectrum contains the most critical information for functional group prediction, which is then effectively refined by the auxiliary inputs.
> > > >
> > > > It is worth emphasizing that the PA-IB framework is a core innovation of our work. To the best of our knowledge, there are no existing studies that utilize this specific asymmetric information bottleneck structure for multi-modal spectroscopic analysis, making direct comparisons with external baselines using this specific ablation setup impossible.
> > > >
> > > >
> > > >
> > > > > *we need to see the author's revised paper to make sure some important changes are made.*
> > > >
> > > > To ensure the updated manuscript thoroughly addresses every point raised by the reviewer, we plan to submit the final revised version after we are confident that all concerns have been resolved through our responses. We believe this strategy will effectively prevent unnecessary back-and-forth exchanges and maximize the respect for your valuable time.  We will proceed to upload the revised manuscript without delay once you are satisfied with the clarifications provided.
> > > >
> > > >
> > > >
> > > > ---------------
> > > >
> > > > We greatly appreciate your insightful and helpful comments, as they will undoubtedly help us improve the quality of our article. If our response has successfully addressed your concerns and clarified any ambiguities, we respectfully hope that you consider raising the score. Should you have any further questions or require additional clarification, we would be delighted to engage in further discussion. Once again, we sincerely appreciate your time and effort in reviewing our manuscript. Your feedback has been invaluable in improving our research.

---

> ### Comment · Reviewer_9qDe · 2025-11-25
> **Overall Response**
>
> `Appreciation.` Overall, thank the authors for giving these rebuttals. For example, the authors have done the dataset distribution analysis and further used micro-f1 and macro-f1 to test the dataset imbalance issues.
>
> `Further questions need to be addressed.`
> - However, there are still some key flaws that need to be solved. I have replied one by one to the rebuttal response. Please see the comments above.
> - Also, we need to see the author's revised paper to make sure some important changes are made. The authors neither revised the paper nor claimed what changes had been down in the rebuttal text, which I think is not a standard professional rebuttal response.
> - Loss of evidence on some of the claims, such as the GPU time estimation, modality choices, and module choices within the pipeline. Please see other detailed comments.
>
> `Ending.` I know the rebuttal stage is short and hard for authors, and we all want to raise our scores based on the authors' hard work; however, I also need to make sure the work's quality and contribution. Please forgive me for giving a 'not good enough' right now. I am happy to see your further comments and response.

---

> ### Author Response · Authors · 2025-11-27
> **Response to Further question for Part II**
>
> > *Can you explain why you said their code is not publicly available? Your baselines in this table are all only graph or SMILES; is there really not a multimodal spectrum model to compare?*
>
>
> Thank you for the opportunity to clarify this point. Regarding DiffSpectra, our earlier wording was imprecise — we meant that although we tried the released implementation, we were unable to reliably reproduce its performance, and the available version is still an early arXiv release. For **SpectraLLM , the work originally cited in your first review
> (https://openreview.net/forum?id=J5XUzUW8o3) does not provide public code**. The repository mentioned this time corresponds to **MolSpectLLM**, which is a different work and was recently withdrawn from ICLR 2026.
>
>
> DiffSpectra, SpectraLLM, MolSpectLLM, and SpectrumWorld indeed represent the latest developments in multi-spectra research. However, the latter three models all adopt LLM backbones, which follow a fundamentally different design philosophy from our lightweight integration framework. It is also important to note that their release dates are extremely close or even later than the ICLR submission deadline (September 24, 2025):
>
> * DiffSpectra — July 9, 2025, Last revised on 5 Nov 2025
>
> * SpectraLLM — August 4, 2025
>
> * MolSpectLLM — September 26, 2025 (after submission)
>
> Given their recency and rapid updates, they were not mature baselines at submission time. Nevertheless, we appreciate the suggestion and will add performance comparisons where feasible to better contextualize our contributions.
>
> **As requested, we compared the performance of MSpecTmol against SpectraLLM and DiffSpectra across two distinct datasets**. Notably, the two baseline models employ divergent definitions of functional groups: SpectraLLM identifies 17 types, whereas DiffSpectra identifies 13. To ensure a fair comparison, we evaluated the Functional Group Similarity[1,2] (FGSim) of our model under each specific definition and compared it directly with the respective baseline. The results are summarized below:
>
>
> | Dataset                  | Evaluation Standard  | Input Modality               | Baseline Model | FGSim  | MspecTmol (Ours) FGSim |
> | :----------------------- | :------------------- | :--------------------------- | :------------- | :----: | :--------------------: |
> | QM9S                     | 17 functional groups | IR                           | SpectraLLM     | 0.6599 |       **0.9328**       |
> |                          |                      | Raman                        | SpectraLLM     | 0.7317 |       **0.9334**       |
> |                          |                      | UV-Vis                       | SpectraLLM     | 0.3713 |       **0.5449**       |
> |                          |                      | IR+Raman+UV-Vis              | SpectraLLM     | 0.7934 |       **0.9781**       |
> | QM9S                     | 13 functional groups | IR                           | DiffSpectra    | 0.9322 |       **0.9501**       |
> |                          |                      | Raman                        | DiffSpectra    | 0.9279 |       **0.9417**       |
> |                          |                      | UV-Vis                       | DiffSpectra    | 0.4354 |       **0.5621**       |
> |                          |                      | IR+Raman+UV-Vis              | DiffSpectra    | 0.9495 |       **0.9830**       |
> | Multimodal Spectroscopic | 17 functional groups | IR                           | SpectraLLM     | 0.6023 |       **0.8095**       |
> |                          |                      | $^{13}$C-NMR                 | SpectraLLM     | 0.4249 |       **0.8039**       |
> |                          |                      | $^{1}$H-NMR                  | SpectraLLM     | 0.3329 |       **0.8124**       |
> |                          |                      | $^{1}$H-NMR+$^{13}$C-NMR     | SpectraLLM     | 0.7209 |       **0.8342**       |
> |                          |                      | $^{1}$H-NMR+$^{13}$C-NMR+ IR | SpectraLLM     | 0.7764 |       **0.8586**       |
>
>
> As shown above, MSpecTmol consistently outperforms both baseline models in both single-spectrum and multi-spectrum configurations, demonstrating the superior capability of our model in extracting chemically meaningful substructures.

---

> > ### Author Response · Authors · 2025-11-27
> > **Response to Further question for Part II**
> >
> > Furthermore, **for the 3D conformation generation task**, the existing multi-spectra LLM-based models do not provide suitable baselines. SpectraLLM and SpectrumWorld do not consider this task at all. MolSpectLLM is a groundbreaking attempt, and it evaluates 3D conformations structure validity and geometry quality. Actually, as far as we are aware, current LLM-based generators frequently fail to produce outputs with consistent atom counts, making the standard RMSD metric undefined and therefore not comparable. This further limits their suitability as direct baselines for our task.
> >
> > DiffSpectra performs direct end-to-end conformation generation, yielding an RMSD around 1.6 Å. In contrast, our method adopts a two-stage paradigm that first reconstructs a molecular graph and then achieve conformations generation, achieving significantly lower RMSD (~ 0.68 Å).  Because the underlying workflows and evaluation settings are not aligned, a direct numerical comparison would be misleading. For these reasons, we believe that including DiffSpectra as a baseline would not provide a fair or meaningful comparison. Nonetheless, we acknowledge its relevance and will mention it in the related work section of the revised manuscript.
> >
> >
> > Since all of the above models are either ICLR submissions from this year or recent arXiv preprints, they were not stable or established enough to be included as baselines at the time of our submission. However,  **we will incorporate a discussion of these works in the Related Work section to clearly position our contributions within this rapidly evolving landscape**.
> >
> >
> >
> > [1] DiffSpectra: Molecular Structure Elucidation from Spectra using Diffusion Models.
> >
> > [2] SpectraLLM: Uncovering the Ability of LLMs for Molecule Structure Elucidation from Multi-Spectra

---

> ### Author Response · Authors · 2025-11-27
> **Response to Further question for Q2**
>
> > *For the Table 5 issue, I found Table 5 is in Appendix G. So it is important that you explicitly reference each table and figure in writing.*
>
>
> Table 5 is presented to demonstrate the trade-off between memory consumption and training time when comparing different numbers of input modalities. The primary goal is to substantiate the conclusion drawn in Appendix G: namely, that substantially increasing the number of modalities incurs unnecessary resource overhead with marginal impact on performance improvement. We will add an appropriate reference to this table in the final version of the manuscript.
>
>
>
> > *Provide the information about model size, batched data size, and so on  for reproducing and checking the training efficiency.*
>
> Regarding your query concerning the training time and memory usage presented in the table, we would like to clarify that **there is no direct or necessary correlation between these two metrics.** To provide adequate reference, we list the specific training parameters used for the 3-modality and 5-modality configurations below:
>
> | **Parameter**                | **3 Modalities (Baseline)** | **5 Modalities (Expanded)** |
> | ---------------------------- | --------------------------- | --------------------------- |
> | **Batch Size**               | 128                         | 128                         |
> | **Learning Rate (lr)**       | 4 × $10^{-4}$          | 4 × $10^{-4}$          |
> | **Model Size (Parameters)**  | $57.4 \text{ M}$            | $90.2 \text{ M}$            |
> | **Epochs**                   | 100                         | 100                         |
> | **Feature Extraction Depth** | 4 (Transformer Layers)      | 6 (Transformer Layers)      |
> | **Initial FC Layer Width**   | 1800                        | 3000                        |
>
> As we state in Appendix B, the experiments were conducted on two A100 (80GB) GPUs. It is important to note that the change in model size is not solely a direct consequence of the increased number of input modalities. Instead, **the model size variation results from experimental adjustments to the network architecture parameters** that we performed to accommodate the increased complexity of the expanded input space. The training time is present in Table 5. By providing these detailed parameters, we hope to fully address your concerns.

---

### Official Review · Reviewer_dotU · 2025-10-31

**Soundness:** 4
**Presentation:** 3
**Contribution:** 3
**Rating:** 8
**Confidence:** 3

**Summary:**

The authors introduce MSpecTmol, a new framework for integrating spectral modalities for identifying molecular structures. Using principles from information bottleneck theory, the authors fuse multiple spectral modalities for improved spectra representations, beating recent baselines in structure identification as well as conformer generation when plugged into a diffusion model.

**Strengths:**

- Results are consistently better than the baselines, sometimes by a large margin. The performance improvements are considerable.
- Good grounding in information bottleneck theory to rationalize their loss functions.
- Architecture is easy to understand an implement, making adoption more feasible.

**Weaknesses:**

- Some minor typographic errors. E.g. "The final objective is: The primary...", "Yu et al. (2022). we could...". A quick overview would suffice.
- Baselines for conformer generation are rather slim. There are a number of recent conformer generation baselines, e.g. [1], [2].
- As the authors admit, the method is restricted to training on a fixed vocabulary of functional groups, limiting performance on novel molecules.

[1] https://arxiv.org/abs/2311.17932
[2] https://arxiv.org/pdf/2507.09785

**Questions:**

- How easily are new modalities added without re-tuning the balancing parameters?
- Have you experimentd with modality-specific priors instead of using a single $q(t_a)$?
- All baselines are machine learning-based. Do you have any comparisons with classical approaches? I am not familiar with this area, but it seems that the method would be more convincing if it outperformed the current a variety of appraoches, not just machine learning models.

---

> ### Author Response · Authors · 2025-11-21
> **Response part I**
>
> We thank the reviewer for recognizing the strong performance, theoretical grounding, and practical architecture of our work. We address each point below.
>
> ---
>
> >  **W1**.*Typographic Errors*
>
> We sincerely apologize for the oversight and will conduct a thorough proofreading pass to correct the identified errors as well as any other typographic issues throughout the manuscript.
>
> ---
>
> >  **W2**.*Conformer Generation Baselines*
>
> We appreciate the reviewer’s mention of recent conformer generation works. However, we task is different from these molecular conformations Generation tasks. We aim to obtain **unique target** molecular conformations that are highly correlated with the molecular spectra, rather than merely generating plausible conformers. Moreover, we wish to clarify that compared to generating conformations directly from 2D molecular structures, **we provide the molecular graph only as a baseline connectivity prior, while the spectra are responsible for delivering stereochemical and spatial constraints for Precise conformation determination**.
>
> To validate the effectiveness of this approach, we conducted a clear and fair comparison with existing graph-only methods. We directly compared our spectrum-conditioned model against baselines that generate conformations solely from graphs, including rule-based methods (RDKit ETKDGv3 and OpenBabel) and deep learning-based methods (GeoDiff). To ensure a fair comparison, we selected the top-5 conformations for each model and calculated the average RMSD for the target conformation.
>
> | Method                      | Condition            | Avg. RMSD (Å) |
> | :-------------------------- | :------------------- | :------------ |
> | RDKit                       | Graph or SMILES      | 1.350         |
> | OpenBabel                   | Graph or SMILES      | 1.279         |
> | GRAPHDG                     | Graph or SMILES      | 1.247         |
> | CGCF                        | Graph or SMILES      | 1.246         |
> | GEOMOL                      | Graph or SMILES      | 1.175         |
> | CONFGF                      | Graph or SMILES      | 1.143         |
> | JODO                        | Graph or SMILES      | 1.124         |
> | GeoDiff                     | Graph or SMILES      | 1.074         |
> | Ours (Single Modality)      | Spectrum + Graph     | 0.697         |
> | **Ours (Three Modalities)** | **Spectrum + Graph** | **0.682**     |
>
> The results indicate that **our model achieves substantially lower RMSD**. This implies that the model can generate accurate molecular conformations directly from additional spectroscopic information, which holds significant value for applications such as drug discovery, protein–ligand interaction analysis, and structure-based drug design.
>
> ----
> >  **W3**: *Limitation to Fixed Functional Group Vocabulary*
>
> We acknowledge that as a supervised classification model, our current output is bound to a predefined vocabulary. However, we emphasize that this design choice is driven by practical utility rather than architectural limitations.
>
> * **Practical Relevance (Appendix F):** As detailed in **Appendix F**, our vocabulary is not arbitrary; the 37 selected functional groups cover the most critical and prevalent structural motifs in organic and medicinal chemistry. This ensures the model's immediate and broad applicability in real-world scenarios, such as drug discovery.
>
> * **Rich Latent Representations (Beyond Classification):** MSpecTmol captures information far beyond this fixed list. Its robust performance on **low-frequency functional groups** and its ability to reconstruct precise 3D structures in the **conformation generation task** (RQ2) serve as strong evidence. These results prove that the model effectively captures rich, multi-level latent representations of spectral information rather than merely memorizing high-frequency tags. This generalized spectrum-structure mapping capability indicates a strong potential to characterize novel chemical entities.
>
> * **Scalability:** The "fixed vocabulary" is a data configuration, not an architectural constraint. Extending MSpecTmol to recognize novel functional groups is a straightforward engineering task: it requires only expanding the label set and fine-tuning the classification head, without necessitating changes to the core fusion framework.

---

> > ### Author Response · Authors · 2025-11-21
> > **Response part II**
> >
> > > **Q1**: *Adding New Modalities Without Re-Tuning Balancing Parameters*
> >
> > Adding new modalities is simple from an architectural perspective, as they are treated as additional auxiliary inputs. The modality embedding can simply be concatenated with the existing auxiliary modality vector, without requiring any modification to the overall framework.
> >
> > Our experiments, presented in **Table 5**, demonstrate consistent performance improvements as more modalities are added (F1: 0.923 → 0.959 → 0.963) without altering hyperparameters. This indicates that the model exhibits **robustness across varying numbers of input modalities** under fixed parameter settings. Therefore, when introducing new modalities, we recommend starting with $\alpha = \beta = 10^{-6}$ and fine-tuning only if performance plateaus. Re-tuning may be beneficial as different modalities possess varying levels of information density, and adjusting $\beta$ helps balance the trade-off between information compression and complementary feature extraction.
> >
> > ---
> >
> > >  **Q2**: *Modality-Specific Priors*
> >
> > We thank the reviewer for this constructive suggestion. In our proposed framework, we originally employed a Gaussian prior for all latent variables, a choice that achieves best performance on both datasets. Following your recommendation to investigate if distinct priors could further enhance the model, we conducted a series of new experiments using modality-specific distributions, with results summarized below:
> >
> > | IR Distribution | ${}^{13}$C-NMR Distribution | ${}^{1}$H-NMR Distribution | F1-Score |
> > | :---: | :---: | :---: | :---: |
> > | Gaussian $\mathcal{N}(0, 1)$ | Gaussian $\mathcal{N}(0, 1)$ | Gaussian $\mathcal{N}(0, 1)$ | **0.959** |
> > | Gamma $\Gamma(1, 1)$ | Gamma $\Gamma(1, 1)$ | Gamma $\Gamma(1, 1)$ | 0.946 |
> > | Laplace $\mathcal{L}(0, 1)$ | Laplace $\mathcal{L}(0, 1)$ | Laplace $\mathcal{L}(0, 1)$ | 0.951 |
> > | Gaussian $\mathcal{N}(0, 1)$ | Uniform $\mathcal{U}[-1, 1]$ | Laplace $\mathcal{L}(0, 1)$ | 0.947 |
> > | Uniform $\mathcal{U}[-1, 1]$ | Gaussian $\mathcal{N}(0, 1)$ | Laplace $\mathcal{L}(0, 1)$ | 0.944 |
> > | Laplace $\mathcal{L}(0, 1)$ | Uniform $\mathcal{U}[-1, 1]$ | Gaussian $\mathcal{N}(0, 1)$ | 0.942 |
> >
> > As shown in the table, employing modality-specific priors **did not improve performance compared to the unified Gaussian baseline** (0.959). This result is well-supported by the Central Limit Theorem, which implies that the aggregation of numerous independent random variables tends toward a normal distribution. Specifically, since the multi-level information in the spectra undergoes extensive processing via multiple MLP layers and self-attention mechanisms prior to compression, the aggregated feature representations naturally tend to approximate a Gaussian distribution. Therefore, we believe that the unified Gaussian prior is the superior choice for aligning with the intrinsic data distribution.
> >
> > ---
> >
> > >  **Q3**: *Comparison with Classical (Non-ML) Approaches*
> >
> > Following your advice, to provide a rigorous benchmark, we compared MSpecTmol against standard non-ML methods using SMILES inputs only. We generated 5 conformations per sample for these baselines and reported the average RMSD:
> >
> > | Method | Condition | Avg. RMSD (Å) |
> > | :--- | :--- | :--- |
> > | RDKit ETKDGv3 | Graph Only | 1.350 |
> > | OpenBabel | Graph Only | 1.279 |
> > | Ours (Single Modality) | Spectrum Only | 0.697 |
> > | **Ours (Three Modalities)** | **Spectrum + Graph** | **0.682** |
> >
> > The results demonstrate that MSpecTmol significantly outperforms non-ML methods. This superior performance is attributed to the rich 3D structural information embedded in spectral data, which offers far greater precision than the heuristic force fields and rule-based algorithms used by standard non-ML approaches.
> >
> > For functional group classification, non-ML methods like **peak matching** and **rule-based systems** have limitations, such as poor scalability and inability to integrate multiple modalities. Our ML-based approach overcomes these challenges by learning generalizable multi-modal rules from large datasets. While we will add a **library search baseline** (e.g., cosine similarity against NIST) for comparison, the strength of ML lies in uncovering subtle correlations between modalities that manual methods cannot capture.
> >
> >
> > -----
> > We greatly appreciate your insightful and helpful comments, as they will undoubtedly help us improve the quality of our article. Should you have any further questions or require additional clarification, we would be delighted to engage in further discussion.

---

> > > ### Comment · Reviewer_dotU · 2025-11-25
> > >
> > > Thanks to the authors for their response and additional experiments. I am further convinced of this paper's quality and recommend its acceptance. I also see the authors have comprehensively answered other reviewers, and I am further assured of its quality. I have increased my confidence to reflect this.
> > >
> > > Best of luck to the authors.

---

> > > > ### Author Response · Authors · 2025-11-27
> > > > **Thank you very much**
> > > >
> > > > Thank you very much for your thoughtful evaluation and for the positive recommendation. We sincerely appreciate your careful reading of our work, your constructive feedback, and your recognition of the additional experiments and clarifications we provided. Your support and encouragement mean a great deal to us, and we are grateful that our efforts helped strengthen the paper. Thank you again for your time and contribution to the review process.
> > > >
> > > > Best.
> > > >
> > > > Authors

---

### Official Review · Reviewer_GdZ5 · 2025-11-01

**Soundness:** 2
**Presentation:** 3
**Contribution:** 2
**Rating:** 2
**Confidence:** 3

**Summary:**

The paper proposes MSrectmol, a multi-modal spectroscopic learning framework designed to integrate information from various spectroscopic modalities, including infrared (IR), nuclear magnetic resonance (NMR), and mass spectrometry (MS), for molecular structure elucidation. The framework introduces a Primary- Auxiliary Information Bottleneck (PA-IB) formulation that extends from the traditional Information Bottleneck to fuse and compress information across different spectroscopic modalities. In this setting, one modality is treated as the primary source of information, while others serve as auxiliary inputs that provide complementary features. The framework also applies adaptive noise gating to fuse and retain task-relevant information across modalities while reducing redundancy.

The framework is applied to two downstream analysis, molecular substructure (functional group) classification and spectrum-conditioned 3D conformation generation. Reported results shown an average micro F1-score of 0.941 across seven different spectrum configurations on the simulated spectrum and 0.866 across six configurations on experimental spectrum. For the conformation generation task, the model attains an average RMSD of 0.695 angstrom across five different input spectrums.
Overall, the methodology appears sound, but it represents an incremental extension of existing information bottleneck and multi-modal fusion methods. The empirical findings partially support the paper’s core contributions.

**Strengths:**

1. Overall, the approach is well motivated and technically sound.
2. The work demonstrates that multi-modal fusion guided by the Primary-Auxiliary Information Bottleneck (PA-IB) can generate representations useful for both functional group identification and 3D conformation prediction.
3. The experiments are conducted across both simulated and experimental datasets.
4. The paper communicates its ideas effectively and meets the clarity and organization standards.

**Weaknesses:**

1.Given that, in conventional practice, chemists first infer 2D molecular connectivity from spectra and then generate 3D structures using cheminformatics tools such as RDKit or OpenBabel, broader benchmarking would strengthen the evaluation of the 3D conformation generation task. The paper includes comparisons with GeoDiff and an attention-based baseline, but additional benchmarks, particularly graph- or SMILES-conditioned conformer generation methods and conventional pipelines using RDKit or OpenBabel, would help contextualize the advantages of direct spectrum-to-3D generation. Such baselines are essential for assessing the reliability and added value of the proposed approach relative to established workflows.

2. The generalization of the model to larger or noisier molecules remains unclear. Current experiments focus on benchmark datasets, leaving open how the framework performs on high-noise experimental spectra or larger molecules with dense peak patterns. Since strong or overlapping spectral peaks often carry critical structural information, additional evaluation on such challenging cases would strengthen the practical validity of the method.

3. The paper does not report computational efficiency metrics such as training cost, inference time, or scalability with increasing modality count or molecular size. Including such results would help assess the framework’s practicality for large-scale spectroscopic applications.

4. Overall, the proposed approach only represents an incremental extension of existing information bottleneck and multi-modal fusion methods. The multi-modal fusion strategy lacks enough novelty.

**Questions:**

1. Could the authors comment on how the model handles noisy or partially missing spectral modalities, which are common in real experimental conditions?
2. The paper compares against GeoDiff and an attention-based baseline. Could the authors provide additional comparisons to a conventional spectrum -> inferred 2D connectivity -> 3D conformer using tools such as RDKit/OpenBabel? Quantitative results from more conformat generation baselines would help evaluate the practical benefit of direct spectrum to 3D generation.
3. Several performance differences relative to baselines appear modest. Could the authors provide statistical tests to indicate which improvements are statistically significant across runs and datasets?
4. While α and β sensitivity are discussed, could the authors provide additional analysis or intuition on how the temperature parameter in the Gumbel–Softmax affects gating sparsity and interpretability?
5. May need to evaluate different fusion strategies.

---

> ### Author Response · Authors · 2025-11-21
> **Response Part I**
>
> We sincerely thank the reviewer for the positive feedback on our motivation, technical soundness, and experimental design. We address each concern below.
>
> ---
>
> >  **W1&Q2**. *Additional benchmarks are essential.*
>
> We appreciate the reviewer’s constructive suggestion regarding the comparison with conventional pipelines. To address this, we have expanded our evaluation to explicitly benchmark against established workflows (including RDKit and OpenBabel) and clarify the advantages of our framework.
>
> **1. Feasibility of Spectrum-to-Structure Inference**
>
> As the reviewer noted, the conventional practice involves inferring 2D connectivity from spectra. Since spectroscopy-to-SMILES prediction is already a well-established direction, we conducted an additional feasibility study by replacing our predictor layer (MLP) with a Transformer-based decoder to predict SMILES strings directly from spectra. **Note that this modification does not alter our underlying PA-IB framework.**
>
> To evaluate performance, we trained the model on the QM9S dataset, which is consistent with the conformation generation task, and calculated the Top-1 and Top-5 accuracy, as well as the MCES score (normalized by the number of heavy atoms). The results are as follows:
>
> | Model          | Top-1 Acc. | Top-5 Acc. | Top-1 mces-score | Top-5 mces-score |
> | :------------- | :--------- | :--------- | :--------------- | :--------------- |
> | Transformer    | 50.02%     | 63.27%     | 0.8674           | 0.9083           |
> | Alberts et al. | 66.59%     | 75.33%     | 0.8821           | 0.9457           |
> | **Ours**       | **70.43%** | **85.25%** | **0.9469**       | **0.9847**       |
>
> These results show that our model achieves high accuracy on the dataset. Even in cases where predictions are not perfectly accurate, the MCES scores close to 1.0 prove that the model can predict molecular graph structures with high similarity. **This demonstrates that spectra contain sufficient structural information to infer molecular identity**, indicating that generating 3D conformations from spectra alone is indeed feasible within our framework.
>
> **2. Comparison with Conventional Graph-Only Pipelines**
> Moreover, we wish to clarify that compared to generating conformations directly from 2D molecular structures, we provide the molecular graph only as a baseline connectivity prior, while the spectra are responsible for delivering stereochemical and spatial constraints that 2D graphs inherently cannot capture, particularly in distinguishing isomers. Based on this "Spectrum-Graph-Conformation" pipeline, we can obtain **molecular conformations that are highly correlated with the molecular spectra**, rather than merely generating plausible conformers that lack a direct correspondence to the spectral information.
>
> To prove the validity of this approach, we conducted a clear and fair comparison with existing graph-only methods as suggested. We directly compared our spectrum-conditioned model with baselines that generate conformations from graphs alone, including rule-based methods (RDKit ETKDGv3 and OpenBabel) and deep learning-based methods (GeoDiff). To ensure a fair comparison, we selected the top-5 conformations for each model and calculated the average RMSD.
>
> | Method                      | Condition            | Avg. RMSD (Å) |
> | :-------------------------- | :------------------- | :------------ |
> | RDKit                       | Graph or SMILES      | 1.350         |
> | OpenBabel                   | Graph or SMILES      | 1.279         |
> | GRAPHDG                     | Graph or SMILES      | 1.247         |
> | CGCF                        | Graph or SMILES      | 1.246         |
> | GEOMOL                      | Graph or SMILES      | 1.175         |
> | CONFGF                      | Graph or SMILES      | 1.143         |
> | JODO                        | Graph or SMILES      | 1.124         |
> | GeoDiff                     | Graph or SMILES      | 1.074         |
> | Ours (Single Modality)      | Spectrum + Graph     | 0.697         |
> | **Ours (Three Modalities)** | **Spectrum + Graph** | **0.682**     |
>
>
> The results indicate that **our model achieves substantially lower RMSD**. This is likely because the baseline models (conventional tools) generate a large pool of potential conformers based on SMILES without a precise target, resulting in larger errors. This demonstrates that spectroscopic information provides additional geometric constraints beyond the molecular graph.
>
> These results collectively demonstrate that spectra alone contain strong structural signals and that our framework is, in principle, capable of operating without molecular graph inputs. At the same time, we must clarify that **the current experimental setup is deliberately designed not for *de novo* structure determination**, but to provide a potential test for rigorously evaluating the quality of the spectral representations learned by our model.

---

> > ### Author Response · Authors · 2025-11-23
> > **Response Part II**
> >
> > >  **W2&Q1**. Generalization to Noisy and Missing Modalities
> >
> > We appreciate the reviewer's insightful suggestion regarding the model's robustness in challenging scenarios. In response, we have added supplementary experiment on larger molecules and consolidated our existing analyses on missing and noisy modalities to demonstrate the generalization capability of MSpecTmol.
> >
> > * **Test on Larger Molecules**
> >
> > To evaluate the model's generalization capability on samples with more complex molecular structures and denser, overlapping spectral peaks, we performed a supplementary stress test on the Alberts et al. dataset. Specifically, instead of a random split, we sorted the entire dataset by heavy atom count. We used the bottom 90% (smaller molecules) for training and the top 10% (largest molecules) for testing. This setup creates a significant distribution shift, requiring the model to infer the structure of complex molecules larger than those seen during training.
> > The results are presented in Table below:
> >
> > | Modality                            | Model          | F1-Score (Larger Molecules, Top 10%) | F1-Score (Original Split) |
> > | :------------ | :------------- | :----------------------------------: | :----------: |
> > | **IR**                              | 1D-CNN         |                0.866                 |           0.895           |
> > |                                     | Transformer    |                0.852                 |           0.881           |
> > |                                     | Wu et al.      |                0.864                 |           0.886           |
> > |                                     | Alberts et al. |                0.874                 |           0.891           |
> > |                                     | **MSpecTmol**  |              **0.900**               |         **0.920**         |
> > | **${}^{13}$C-NMR**                  | 1D-CNN         |                0.623                 |           0.674           |
> > |                                     | Transformer    |                0.845                 |           0.913           |
> > |                                     | Wu et al.      |                0.873                 |           0.914           |
> > |                                     | Alberts et al. |                0.896                 |           0.919           |
> > |                                     | **MSpecTmol**  |              **0.904**               |         **0.923**         |
> > | IR + ${}^{13}$C-NMR + ${}^{1}$H-NMR | 1D-CNN         |                0.873                 |           0.900           |
> > |                                     | Transformer    |                0.902                 |           0.936           |
> > |                                     | Wu et al.      |                0.912                 |           0.944           |
> > |                                     | Alberts et al. |                0.916                 |           0.947           |
> > |                                     | **MSpecTmol**  |              **0.925**               |         **0.959**         |
> >
> >
> >
> >
> > As shown in Table, while the performance on unseen larger molecules naturally dips compared to the standard random split, MSpecTmol (Three modalities) maintains a high F1-score of 0.925. This demonstrates that our PA-IB framework effectively learns intrinsic spectroscopic-structural correlations rather than simply memorizing dataset-specific patterns, confirming its capability to generalize to more complex chemical spaces.
> >
> > *  **Test on  Missing Modalities**
> >
> > We have rigorously evaluated the model's robustness against incomplete data which is a common real-world challenge in Appendix M and Figure 10(b) of our paper. In this experiment, we masked individual spectral modalities in the test set to assess performance degradation.
> >
> > | Input Configuration            | MSpecTmol | 1D-CNN | Transformer | Wu et al. | Alberts et al. |
> > | :--------------- | :-------- | :----- | :---------- | :-------- | :------------- |
> > | MS+ $^{13}$C-NMR + $^{1}$H-NMR | 0.913     | 0.847  | 0.858       | 0.872     | 0.881          |
> > | w/o MS                         | 0.862     | 0.811  | 0.826       | 0.831     | 0.847          |
> > | w/o $^{1}$H-NMR                | 0.897     | 0.823  | 0.834       | 0.848     | 0.866          |
> > | w/o $^{13}$C-NMR               | 0.878     | 0.819  | 0.818       | 0.832     | 0.851          |
> >
> >
> > As shown in Table, the F1-score drops from 0.9134 to scores between 0.8623 and 0.8974 when a modality is absent. This minimal degradation indicates that MSpecTmol does not fail catastrophically when data is incomplete; instead, it leverages the primary-auxiliary framework to effectively supplement missing information from the remaining available spectra.

---

> ### Author Response · Authors · 2025-11-23
> **Response Part III**
>
> *  **Robustness to Spectral Noise**
> The model's performance under noisy conditions is reported in **Appendix M** and Figure 10(c). We simulated real-world experimental noise by injecting Gaussian noise into the test spectra, dynamically scaled relative to the maximum peak intensity.
>
> | Noise Level ($\sigma$) | MSpecTmol | 1D-CNN | Transformer | Wu et al. | Alberts et al. |
> | :--------------------- | :-------- | :----- | :---------- | :-------- | :------------- |
> | 0.00                   | 0.9134    | 0.847  | 0.858       | 0.872     | 0.881          |
> | 0.02                   | 0.8820    | 0.815  | 0.825       | 0.838     | 0.846          |
> | 0.05                   | 0.8250    | 0.745  | 0.761       | 0.767     | 0.772          |
> | 0.10                   | 0.7469    | 0.635  | 0.704       | 0.711     | 0.703          |
>
>
> As illustrated in Table, even under significant noise perturbation, the model maintains a respectable F1-score of 0.7469. This proves that our information bottleneck mechanism successfully filters out irrelevant noisy features while preserving core structural information.
>
> >  **W3**. Reporting computational efficiency metrics.
>
> We thank the reviewer for the suggestion. Evaluating computational efficiency is crucial for assessing the framework's practicality. We have conducted a comprehensive analysis of training costs, inference latency, and scalability.
>
> * **Parameters**
>
> We evaluated the training efficiency of MSpecTmol compared to baseline models. All models were trained on **two NVIDIA A800 (80GB) GPUs**. As shown in the table below,  Transformer has the least parameters, approximately 30.4M, but its training time needs 35 hours as shown in Table 4. And the parameter quantities of other baselines are comparable. MSpecTmol achieves the highest F1-score (0.959) with a moderate training time of 2.5 hours.
>
> | Model          | Parameters (MB) | F1-score  |
> | :------------- | :-------------- | :-------- |
> | 1D-CNN         | 47.8            | 0.900     |
> | Transformer    | 30.4            | 0.936     |
> | Wu et al.      | 52.1            | 0.944     |
> | Alberts et al. | 55.4            | 0.947     |
> | **MSpecTmol**  | **57.4**        | **0.959** |
>
>
> * **Inference Time**
>
> To assess the model's suitability for high-throughput screening, we measured the total inference time for processing **10,000 samples** on a single **NVIDIA A100 GPU**. The table below shows that MSpecTmol completes the task in just 14.0 seconds. This speed is comparable to the simple 1D-CNN (10.4 s) and drastically faster than the Transformer (45.1s), confirming its efficiency for real-time applications.
>
> | Model          | Inference Time (s) |
> | :------------- | :----------------- |
> | 1D-CNN         | 10.4               |
> | Transformer    | 45.1               |
> | Wu et al.      | 55.4               |
> | Alberts et al. | 60.1               |
> | **MSpecTmol**  | **14.0**           |
>
> * **Scalability with Increasing Modality Count**
>
> The following table shows how the performance and resource consumption scale with the number of input modalities. As expected, adding modalities increases memory usage and training time. The transition from 1 to 3 modalities yields a significant performance gain (F1: 0.923 $\to$ 0.959) with manageable cost. However, extending to 5 modalities nearly doubles the computational cost for only a marginal performance improvement. Thus, the 3-modality configuration provides the optimal balance between efficiency and accuracy.
>
> | Modalities | Memory (GB) | Training Time (h) | F1-score  |
> | :--------- | :---------- | :---------------- | :-------- |
> | 1          | 3.4         | 2.0               | 0.923     |
> | 3          | 6.6         | 2.5               | **0.959** |
> | 5          | 11.3        | 5.0               | 0.963     |
>
> * **Scalability with Molecular Size**
>
> Finally, to verify whether the model's computational cost is sensitive to molecular complexity, we measured the training time on subsets of data sorted by **Heavy Atom Count** (using consistent subsets of 100,000 samples). As presented in the table below, the training time remains remarkably consistent (~560s) across different molecular sizes. This is because our model takes fixed-dimension interpolated spectra as input and outputs functional group probabilities; consequently, the physical size or complexity of the molecule does not alter the input tensor dimensions or the model architecture.
>
> | Heavy Atom Count | Training Time (s) |
> | :--------------- | :---------------- |
> | 5 - 15           | 564               |
> | 16 - 25          | 556               |
> | 25 - 35          | 558               |

---

> > ### Author Response · Authors · 2025-11-23
> > **Response Part IV**
> >
> > >  **W4**. Novelty of Multi-Modal Fusion
> >
> >
> > We appreciate the reviewer’s comment. We would like to clarify that traditional information fusion methods are typically based on **symmetric information fusion paradigms** (e.g., simple concatenation or standard Multi-view IB), where modalities are treated equally. In contrast, our PA-IB innovatively employs an **asymmetric, conditional information bottleneck** approach to strictly address the limitations of these symmetric formulations.
> >
> > In the context of spectral analysis, distinct spectra (such as IR, MS, and NMR) encode information at different structural levels; they contain unique insights but also share substantial overlapping information. Traditional methods typically treat these modalities as independent or equivalent sources. This approach inevitably leads to **information redundancy**, causing critical, distinctive signals to be "drowned out" or overshadowed by the repetitive information shared across modalities.
> >
> > To address this, our framework innovatively utilizes a Primary-Auxiliary architecture. Specifically, our method differs in two key aspects:
> >
> > 1.  **Conditional Objective:** We design a conditional IB term, $-I(Y; T_a | T_m)$, which ensures that the auxiliary spectra contribute only novel, complementary information that the primary modality has not captured, rather than duplicating existing features.
> >
> > 2.  **Redundancy Penalization:** We introduce a regularization term, $\beta I(T_a; X_m, X_a)$, to explicitly reduce the redundancy between primary and auxiliary inputs, whereas traditional IB methods penalize the complexity of each modality independently.
> >
> > To validate its effectiveness, we compared PA-IB against standard Early, Mid-level, and Late fusion strategies. **Referring to the extra experimental results detailed in Q5**, MSpecTmol consistently outperforms all three baselines. This superior performance demonstrates that while conventional fusion methods often suffer from redundancy or ignore inter-modal interactions (as seen in the lower performance of Early/Mid-level and Late fusion), our asymmetric conditional framework successfully synthesizes heterogeneous spectral data by filtering noise and maximizing complementary information gain.
> >
> > In summary, the core novelty of PA-IB lies in its tailored asymmetric mechanism for heterogeneous spectroscopic data, offering a principled solution to the redundancy problem that standard symmetric fusion methods fail to address.
> >
> > >  **Q3**. Statistical Significance.
> >
> > Follow your request, we have rerun the experiments 10 times and conducted a detailed performance comparison between our model and the second-best performing model, calculating the corresponding $P$-values. The results are presented in the tables below:
> >
> > **QM9S Dataset (Simulated Spectra)**
> >
> > | Spectrum Config          | Second-best. | MSpecTmol | P-value |
> > | :----------------------- | :----------- | :-------- | ------: |
> > | IR                       | 0.891        | 0.923     |  1.3e-4 |
> > | $^{13}$C-NMR             | 0.919        | 0.920     |  2.5e-3 |
> > | $^{1}$H-NMR              | 0.946        | 0.927     |  7.9e-5 |
> > | IR + ($^{13}$C, $^{1}$H) | 0.947        | 0.959     |  4.1e-3 |
> > | $^{13}$C + (IR, $^{1}$H) | 0.947        | 0.957     |  8.8e-4 |
> > | $^{1}$H + (IR, $^{13}$C) | 0.947        | 0.956     |  3.3e-3 |
> > | IR + (MS/MS)             | 0.931        | 0.944     |  5.5e-5 |
> >
> > **SDBS Dataset (Experimental Spectra)**
> >
> > | Spectrum Config          | Second-best. | MSpecTmol | P-value |
> > | :----------------------- | :----------- | :-------- | ------: |
> > | MS                       | 0.836        | 0.847     |  1.8e-3 |
> > | $^{13}$C-NMR             | 0.836        | 0.842     |  9.2e-4 |
> > | 1H-NMR                   | 0.803        | 0.792     |  2.7e-5 |
> > | MS + ($^{13}$C, $^{1}$H) | 0.881        | 0.913     |  4.6e-5 |
> > | $^{13}$C + (MS, $^{1}$H) | 0.881        | 0.894     |  5.1e-3 |
> > | $^{1}$H + (MS, $^{13}$C) | 0.881        | 0.909     |  7.4e-5 |
> >
> > The results indicate that the $P$-values for the comparison between our model and the second-best model are consistently below $1 \times 10^{-3}$ (or $1\text{e-}3$). This conclusively demonstrates the **statistical significance** of our model's superior performance. We will add this result in revised version.

---

> > > ### Author Response · Authors · 2025-11-23
> > > **Response Part V**
> > >
> > > >  **Q4**.Gumbel-Softmax temperature analysis
> > >
> > > We thank the reviewer for the suggestion and conducted a sensitivity analysis across temperatures (t $\in$ {0.5, 1.0, 1.5, 2.0}):
> > >
> > > | Temperature ((t)) | F1-score  |
> > > | ----------------- | --------- |
> > > | 0.5               | 0.952     |
> > > | **1.0**           | **0.959** |
> > > | 1.5               | 0.954     |
> > > | 2.0               | 0.945     |
> > >
> > > As shown above, when the temperature is set to a **low value** ((t = 0.5)), the model performance slightly drops. At the **optimal temperature** ((t = 1.0)), we obtain the best F1-score. When the temperature becomes **higher** ((t = 1.5, 2.0)), the performance degrades more noticeably. We will include this analysis in the Appendix.
> > >
> > > Intuitively, the temperature controls the **sparsity and sharpness of the gating** over spectral frequency bands. In our setting, each gate decides whether a local region of each spectrum is preserved or replaced by noise; very low (t) makes these decisions almost binary, which can discard weak but chemically informative peaks (e.g., small shoulders or minor bands) that are important for distinguishing functional groups and isomers. Conversely, high (t) yields overly soft gates, causing most bands to be partially retained, which weakens redundancy suppression and blurs the importance patterns across modalities. The best performance at (t = 1.0) is therefore consistent with PA-IB’s goal of learning **selective yet stable masks** that retain structurally informative spectral regions while filtering out redundant or noisy segments.
> > >
> > >
> > >
> > > > **Q5**. Evaluating different fusion strategies
> > >
> > > PA-IB is a multi-modal fusion strategy that combines Information Bottleneck (IB), aimed at **removing redundant information through asymmetric information compression and extracting discriminative information** from auxiliary modalities to complement the features of the primary modality.  To validate its effectiveness as a multi-modal information fusion strategy, we conducted an ablation study comparing PA-IB with models that use three different information fusion strategies:
> > >
> > > * **Early Fusion**: The features from all three modalities are directly concatenated before being input into the model, allowing the model to learn a joint representation from the raw data.
> > > * **Mid-level Fusion**: Each modality is processed through CNNs to extract features, which are then concatenated and used as input features for an MLP.
> > > * **Late Fusion**: Each modality independently predicts the presence of functional groups through an MLP, and the final prediction is obtained by averaging the predictions from all modalities.
> > >
> > > | Modality                      | MSpecTmol | Early Fusion | Mid-level Fusion | Late Fusion |
> > > | ----------------------------- | --------- | ------------ | ---------------- | ----------- |
> > > | IR + $^{13}$C-NMR + $^1$H-NMR | 0.959     | 0.900        | 0.904            | 0.874       |
> > > | IR + MS/MS_pos + MS/MS_neg    | 0.944     | 0.887        | 0.895            | 0.854       |
> > >
> > > From the results, MSpecTmol superior three fusion baselines (F1-score index). Early fusion and mid-level fusion perform better than late fusion, likely because they provide some level of joint representation learning but still fail to remove redundant cross-modal information. Late fusion performs the worst since it ignores inter-modal interactions entirely, processing each modality independently and merely averaging predictions, which leads to substantial information loss.

---

> > > > ### Author Response · Authors · 2025-11-27
> > > >
> > > > Comment: Dear Reviewer,
> > > >
> > > > We sincerely thank you for your thoughtful insights provided in your review. We deeply appreciate the time and effort you dedicated to improving our work. We would be grateful if you could let us know whether our revisions and responses adequately address your concerns, or if there are any remaining points we can clarify.
> > > >
> > > > Best, The authors

---

### Official Review · Reviewer_c4BW · 2025-11-01

**Soundness:** 3
**Presentation:** 3
**Contribution:** 2
**Rating:** 2
**Confidence:** 4

**Summary:**

The paper proposes a unified multi-modal approach for molecular structure elucidation from spectroscopic data.

The authors introduce a Primary–Auxiliary Information Bottleneck (PA-IB) framework to model complementary information across multiple spectra (IR, NMR, MS, etc.). They evaluate the method on two main tasks: (1) functional-group classification (structure elucidation), and (2) 3D conformation generation. Experiments reportedly show superior performance compared to several CNN and Transformer baselines.

**Strengths:**

1. The paper addresses a relevant question: how to fuse information from multiple spectroscopy modalities for molecular structure understanding.

2. The use of information bottleneck principles for spectral representation compression is theoretically sound and aligns with prior probabilistic learning methods.

3. The multi-modal fusion experiments for functional-group prediction are relatively well-executed, and the analysis of modality importance (IR, H-NMR, C-NMR) provides useful insight.

**Weaknesses:**

1. The novelty of the proposed method is rather limited, as the PA-IB framework represents a straightforward extension of the conventional information bottleneck to a multi-modal context. The architectural choices, such as 1D-CNN encoders, follow common practice and do not introduce a fundamentally new modeling mechanism.
2. The experimental design for the conformation generation task is scientifically inconsistent with practice and the paper’s stated goal of structure elucidation from spectra. As described in Appendix N.1, the generation model is conditioned on the complete molecular graph, including atom types and bond connections, which are precisely the unknowns that spectroscopy is supposed to infer.
3. The paper lacks an ablation to verify the contribution of the primary–auxiliary design within the PA-IB formulaton. The improvement might simply come from adding information-bottleneck regularization rather than from the asymmetric modality treatment. An additional experiment applying a uniform IB constraint across all modalities would clarify whether the proposed hierarchical design truly provides unique benefits.

**Questions:**

Please refer to the weaknesses above.

---

> ### Author Response · Authors · 2025-11-21
> **Response Part I**
>
> We sincerely thank the reviewer for the thoughtful feedback and for recognizing the relevance of our work in multi-modal spectroscopic fusion. Below, we have carefully considered and responded to your valuable comments point by point.
>
> ---
>
> > **W1**. *Limited Novelty of PA-IB Framework*
>
> We appreciate the reviewer’s insightful comment and apologize for not clearly conveying the conceptual novelty of our method. Our contribution goes beyond simply extending the Information Bottleneck (IB) framework to a multi-modal setting. The key innovation lies in establishing an asymmetric information fusion paradigm with a principled conditional IB formulation, for the inherent asymmetry of multi-modal spectroscopic data for molecular structure identify.
>
> Conventional multi-modal IB approaches typically treat all modalities as exchangeable, applying symmetric and unconditional compression to each latent representation. However, this assumption is inconsistent with spectroscopic data, where IR, NMR, and MS encode distinct and complementary physical constraints on molecular structure (Barone et al., 2021), thus requiring a specialized mechanism for complementary information extraction. Our proposed PA-IB introduces a directional primary–auxiliary architecture to achieve this.
>
> * (i) We design a conditional IB term, $-I(Y; T_a | T_m)$, ensuring that auxiliary spectra contribute novel and complementary information. Unlike the classical IB objective, which uniformly maximizes $I(Y; T)$ across modalities, PA-IB encourages auxiliary channels to enrich rather than duplicate the representation learned from the primary modality.
> * (ii) We further introduce a regularization term, $\beta I(T_a; X_m, X_a)$, which explicitly reduces redundancy between primary and auxiliary spectra, whereas traditional IB penalizes $I(T; X)$ independently for each modality.
>
> As shown in Table 1, while simple CNN-based fusion models achieve only a marginal improvement (from 0.895 to 0.900 when adding modalities), our PA-IB framework yields a substantial performance gain (from 0.923 to 0.959). This significant improvement validates the effectiveness of PA-IB in discarding redundant multi-modal information while preserving discriminative features, leading to a more powerful and efficient model for molecular structure identification.
>
> * Barone V, Alessandrini S, Biczysko M, et al. Computational molecular spectroscopy[J]. _Nature Reviews Methods Primers_, 2021, 1(1): 38.
>
> > **W2**. *Conformation Generation Task Design*
>
> We appreciate the reviewer’s insightful comment. Our current implementation indeed conditions the diffusion model on both the molecular graph and the spectra. However, this design choice should not be interpreted as a conceptual limitation of our framework. Conditioning on the molecular graph primarily serves to stabilize training and improve geometric fidelity, especially in early diffusion steps.
>
> **Importantly, it is also technically feasible to generate complete 3D conformations from spectra directly.** Since spectroscopy-to-SMILES prediction is already a well-established research direction, we conducted an additional feasibility study by replacing our predictor layer (MLP) with a Transformer-based decoder to predict SMILES strings directly from spectra. **This modification does not alter our underlying PA-IB framework.** To evaluate performance, we trained the model on the QM9S dataset—consistent with the conformation generation task—and calculated the Top-1 and Top-5 accuracy of the generated results, as well as the MCES score (normalized by the number of heavy atoms). The results are as follows:
>
>
> | Model          | Top-1 Acc. | Top-5 Acc. | Top-1 mces-score | Top-5 mces-score |
> | :------------- | :--------- | :--------- | :--------------- | :--------------- |
> | Transformer    | 50.02%     | 63.27%     | 0.8674           | 0.9083           |
> | Alberts et al. | 66.59%     | 75.33%     | 0.8821           | 0.9457           |
> | **Ours**       | **70.43%** | **85.25%** | **0.9469**       | **0.9847**       |

---

> ### Author Response · Authors · 2025-11-21
> **Response Part II**
>
> These results show that our model achieves high accuracy on the dataset. Even in cases where predictions are not perfectly accurate, the MCES scores close to 1.0 demonstrate that the model is capable of reconstructing molecular graph structures with high similarity. This demonstrates that spectra contain sufficient structural information to infer molecular identity, indicating that generating 3D conformations from spectra alone is indeed feasible within our framework.
>
> Moreover, we wish to clarify that compared to generating conformations directly from 2D molecular structures, **we provide the molecular graph only as a baseline connectivity prior, while the spectra are responsible for delivering stereochemical and spatial constraints for Precise conformation determination**. Based on this pipeline, we can obtain **unique target** molecular conformations that are highly correlated with the molecular spectra, rather than merely generating plausible conformers.
>
> To validate the effectiveness of this approach, we conducted a clear and fair comparison with existing graph-only methods. We directly compared our spectrum-conditioned model against baselines that generate conformations solely from graphs, including rule-based methods (RDKit ETKDGv3 and OpenBabel) and deep learning-based methods (GeoDiff). To ensure a fair comparison, we selected the top-5 conformations for each model and calculated the average RMSD for the target conformation.
>
> | Method                      | Avg. RMSD (Å) |
> | :-------------------------- | :------------ |
> | RDKit                       | 1.350         |
> | OpenBabel                   | 1.279         |
> | GRAPHDG                     | 1.247         |
> | CGCF                        | 1.246         |
> | GEOMOL                      | 1.175         |
> | CONFGF                      | 1.143         |
> | JODO                        | 1.124         |
> | GeoDiff                     | 1.074         |
> | Ours (Single Modality)      | 0.697         |
> | **Ours (Three Modalities)** | **0.682**     |
>
> The results indicate that **our model achieves substantially lower RMSD**. This improvement likely stems from the fact that baseline models generate a large pool of potential conformers based on SMILES or 2D graph without a precise target, resulting in larger errors. This demonstrates that spectroscopic information is needed to provides additional geometric constraints.
>
> These results collectively demonstrate that spectra alone contain strong structural signals and that our framework is, in principle, capable of operating without molecular graph inputs. At the same time, we must clarify that **the current experimental setup is deliberately designed not for *de novo* structure determination**, but rather as a rigorous test to evaluate the quality of the spectral representations learned by our model.
>
> > **W3**. *Ablation Study: Uniform IB vs. PA-IB**
>
> Thank you for this insightful suggestion. We verify the contribution of the primary–auxiliary design from two complementary perspectives.
>
> First, Table 1 in the main paper shows results using each modality as the primary one while treating the remaining two as auxiliary. We observe clear performance variation: selecting IR (the most informative modality for functional-group presence) as the primary yields the best results. This supports the importance of assigning a **dominant modality** and enhancing it using auxiliary spectra—an effect that cannot be captured by symmetric multimodal IB.
>
> Second, following your recommendation, we conducted a direct ablation comparing our PA-IB formulation to a **symmetric baseline** that applies uniform IB constraints to all modalities:
>
> * **PA-IB (Ours):**
>   $(-I(Y;T_m) - I(Y;T_a \mid T_m) + \alpha I(X_m;T_m) + \beta I(T_a; X_m, X_a))$
>
> * **Uniform IB:**
>   $(-I(Y;T_m, T_a) + \alpha I(X_m;T_m) + \beta I(X_a;T_a))$
>
> The Uniform IB baseline independently compresses each modality without modeling complementary information or cross-modal redundancy.
>
> | Objective        | IR        | $^{13}C$-NMR | $^1H$-NMR | IR + $^{13}C$-NMR + $^1H$-NMR |
> | ---------------- | --------- | -------------- | ------------- | --------------------------- |
> | **PA-IB (Ours)** | **0.923** | **0.920**      | **0.927**     | **0.959**                   |
> | Uniform IB       | 0.906     | 0.904          | 0.905         | 0.934                       |
>
> PA-IB outperforms Uniform IB across all settings, with a **2.5% absolute gain** in the multimodal case (0.959 vs. 0.934). Uniform IB suffers from redundant cross-modal information, confirming that symmetric treatment is suboptimal. The performance gap widens in the multimodal setting, demonstrating that PA-IB’s superiority.
>
> ---
> We greatly appreciate your insightful and helpful comments, as they will undoubtedly help us improve the quality of our article. If our response has successfully addressed your concerns and clarified any ambiguities, we respectfully hope that you consider raising the score.

---

> > ### Comment · Reviewer_c4BW · 2025-11-25
> >
> > Thank you for the detailed response. The additional clarifications and ablation results have addressed my main concerns, especially the ablation comparing PA-IB and uniform IB, which supports the necessity of the proposed asymmetric information bottleneck design. According to the authors’ response, I am willing to raise my rating.

---

### Author Response · Authors · 2025-12-01
**Author Final Remarks by Authors**

Dear AC:

Thank you very much for your time, effort, and dedicated service. Although our paper initially received relatively low scores, we have carefully addressed the reviewers’ major concerns through extensive additional experiments and detailed clarifications. The reviewers were rigorous and thorough, and **through step-by-step explanations of our contributions, along with the supplementary results they requested**, we gradually gained their understanding and positive acknowledgment. **This progress has been hard-won for us**, and we sincerely appreciate the constructive nature of the review process.

Next, we would like to summarize the key pros and cons noted by the reviewers, along with our responses, for your convenience:

> **Key pros noted by the reviewers:**

**S1.** The core idea is well-motivated, and theoretically grounded. (`All Reviewers`)

**S2.** The experimental evaluation is comprehensive with strong performance (`All Reviewers`)

**S3.** The visualization provide strong interpretability and practical insights (`Reviewers c4BW, GdZ5, dotU`)

**S4.** The paper is well-presented and easy to follow (`Reviewers c4BW, GdZ5, dotU`)

> **Key cons noted by the reviewers:**

`Reviewer c4BW` primarily focused on the novelty of our PA-IB formulation relative to conventional Information Bottleneck theory, and the lack of certain baseline comparisons and ablation experiments. In our response, we provided a detailed clarification of how PA-IB differs conceptually and operationally from the classical IB framework, and we also added the requested experiments and ablation results. After reviewing our response results, `Reviewer c4BW` **understanded the novelty of PA-IB and improve to a positive score**.

----
`Reviewer GdZ5` requested additional generative-model baselines, a more thorough generalization analysis, and a detailed report on computational efficiency. In our response, we provided comprehensive explanations and substantially expanded the experimental section including additional benchmarks, extended generalization analyses, and temperature hyperparameter studies, with a detailed efficiency evaluation. These additions fully address the concerns raised. **As of now, the reviewer has not yet responded, but** `Reviewer GdZ5's` **concerns are similar to**`Reviewer 9qDe`.

---
`Reviewer dotU` suggested adding additional baseline models and correcting typo errors, and also raised concerns regarding the method’s potential restriction to a fixed vocabulary. During the response phase, we provided comprehensive clarifications, including a detailed explanation of the relationship between our framework and the notion of a “fixed vocabulary,” and demonstrated why this does not limit the applicability of our approach. **These detailed responses effectively addressed the reviewer’s concerns, and** `Reviewer dotU` **gave a fully positive attitude.**

---
`Reviewer 9qDe` raised concerns regarding our motivation, the comprehensiveness of the evaluation metrics, and the effectiveness of the primary–auxiliary encoding module, and requested additional training details and ablation studies (**Round I**). We provided thorough and detailed responses, which **resolved several of these concerns**.

**In Round II**, the reviewer questioned why models such as DiffSpectra, SpectraLLM, and MolSpectLLM were not included as baselines, and asked for more training details. We clarified that these models were submissions to ICLR 2026 in the same time and were not available at our initial submission. Furthermore, these works adopt LLM-based architectures, which differ from our CNN+diffusion framework, making them unsuitable as direct baselines. Nevertheless, we **still provided the requested comparative results in the rebuttal** and explained why they could not be incorporated into the main paper due to their unpublished status.

Meanwhile, we supplied comprehensive hyperparameters and training configurations to fully address the reviewer’s concerns. Throughout the exchange, we engaged in constructive discussion, and `Reviewer 9qDe` **explicitly expressed satisfaction, stating: “I am happy to see your further comments and response.”**

---

Due to reviewing policy constraints, we could not receive further comments from the reviewer. We respectfully ask the AC to consider the full rebuttal exchange, which shows that reviewers’ concerns were substantively addressed and acquired positively acknowledged through extensive experiments and clarifications.

---
In summary, inferring molecular substructures or even 3D spatial information from multi-modal spectroscopy is a relatively new direction with limited prior work. **Some initial concerns arose from misunderstandings, but our detailed clarifications and extensive additional experiments have resolved these issues and demonstrated the validity and acquired positively acknowledged**. Finally, we sincerely thank the AC for the time and patience invested in this process.

Best

Authors

---

### Note · Authors · 2026-01-29

I have read and agree with the venue's withdrawal policy on behalf of myself and my co-authors.

---

### Meta-Review · Area_Chair_epEa · 2026-01-04

**Summary:**

This work addresses a practically relevant problem of fusing multiple spectroscopic modalities for molecular structure understanding. The approach is built on information bottleneck theory with a clear theoretical motivation and a well-structured technical framework. The multimodal fusion experiments achieve consistently better performance than the baselines on functional group prediction and conformation generation tasks, and the analysis of modality importance provides useful chemical insights. The paper is generally well organized, the methodology is easy to follow, and the implementation appears reproducible.
The reviewers pointed out several issues with this paper: the overall novelty is limited; the experimental design of the conformation generation task is scientifically inconsistent with typical spectroscopic structure elucidation pipelines; moreover, the ablation studies and baseline comparisons are insufficient to clearly disentangle the contributions of the primary–auxiliary module design and individual components. In addition, the generalization of the method to noisy or partially missing spectra, larger molecules, and its computational efficiency in practical scenarios has not been sufficiently explored.

**Reviewer Concerns:**

The ablation study comparing PA-IB with uniform IB provides a sufficient and convincing response from the authors, effectively addressing Reviewer c4BW’s concern. The concerns raised by Reviewer dotU have also been well addressed by the authors. Reviewer 9qDe raised comprehensive concerns regarding both the motivation and the experimental evaluation. The authors responded actively, and the overall response is relatively thorough.
In response to Reviewer GdZ5’s comments, the authors have addressed the raised points in detail. However, two issues remain. First, the novelty of the paper is still a concern. Second, the authors need to explain how the model handles missing modalities at a methodological level, rather than only evaluating its performance under missing-modality settings. These aspects require further consideration by the authors.

**Reviewer Scores:**

Reviewer c4BW is expected to increase their score, as the authors have addressed the concerns raised through additional experimental results. Reviewer dotU is likely to maintain the current acceptance recommendation. Reviewer 9qDe has engaged in in-depth discussions with the authors, and I believe they are also likely to raise their score. However, based on the current reviewer feedback, the primary concern remains the novelty of the paper. Even if some reviewers increase their scores, the work may still fall short of the principled contributions expected by ICLR, and further deep reflection and innovation from the authors are needed.

---

### Decision · Program_Chairs · 2026-01-26

Reject